# Epigenetic modifications regulate cultivar-specific root development and metabolic adaptation to nitrogen availability in wheat

Hao Zhang[1,2], Zhiyuan Jin[3,4], Fa Cui[5], Long Zhao[1,2], Xiaoyu Zhang[1,2], Jinchao Chen[1,2], Jing Zhang[1,2], Yanyan Li[1,6], Yongpeng Li[6], Yanxiao Niu[3,4], Wenli Zhang [7], Caixia Gao [1,2], Xiangdong Fu [1,2], Yiping Tong[1,2], Lei Wang[1,6], Hong-Qing Ling [1,2,8] ✉, Junming Li[4,5,6] ✉ & Jun Xiao [1,2,9] ✉

The breeding of crops with improved nitrogen use efficiency (NUE) is crucial for sustainable agriculture, but the involvement of epigenetic modifications remains unexplored. Here, we analyze the chromatin landscapes of two wheat cultivars (KN9204 and J411) that differ in NUE under varied nitrogen conditions. The expression of nitrogen metabolism genes is closely linked to variation in histone modification instead of differences in DNA sequence. Epigenetic modifications exhibit clear cultivar-specificity, which likely contributes to distinct agronomic traits. Additionally, low nitrogen (LN) induces H3K27ac and H3K27me3 to significantly enhance root growth in KN9204, while remarkably inducing *NRT2* in J411. Evidence from histone deacetylase inhibitor treatment and transgenic plants with loss function of H3K27me3 methyltransferase shows that changes in epigenetic modifications could alter the strategy preference for root development or nitrogen uptake in response to LN. Here, we show the importance of epigenetic regulation in mediating cultivar-specific adaptation to LN in wheat.

Improving nitrogen-use efficiency (NUE) has emerged as a pressing requirement for sustainable agriculture[1]. NUE is mainly composed of two components: N uptake efficiency (NUpE) and N utilize efficiency (NUtE). NUpE pertains to the acquisition of nitrogen from the soil, while NUtE signifies the yield generated per unit of nitrogen obtained. In situations where nitrogen availability is limited for crops such as winter wheat[2], barley[3], studies indicate that NUpE holds greater importance in determining NUE compared to NUtE. To enhance both

NUE and grain yield, specifically under low nitrogen conditions, it is imperative to boost nitrogen uptake, which is predominantly regulated by nitrate transporters. The concentration of nitrate in soil can experience substantial fluctuations due to external environmental factors; as a result, root system architecture plays a vital role in the efficient absorption of nitrate. Numerous efforts have been undertaken to assist plants in acquiring adequate nitrogen under such conditions, such as overexpressing *NRT2* (a high affinity nitrogen

[1]State Key Laboratory of Plant Cell and Chromosome Engineering, Institute of Genetics and Developmental Biology, Chinese Academy of Sciences, Beijing 100101, China. [2]University of Chinese Academy of Sciences, Beijing 100049, China. [3]Ministry of Education Key Laboratory of Molecular and Cellular Biology, Hebei Research Center of the Basic Discipline of Cell Biology, Hebei Key Laboratory of Molecular and Cellular Biology, Hebei Normal University, Shijiazhuang 050024, China. [4]Hebei Collaboration Innovation Center for Cell Signaling, Shijiazhuang 050024, China. [5]Key Laboratory of Molecular Module-Based Breeding of High Yield and Abiotic Resistant Plants in Universities of Shandong, College of Agriculture, Ludong University, Yantai 264025, China. [6]Center for Agricultural Resources Research, Institute of Genetics and Developmental Biology, Chinese Academy of Sciences, Shijiazhuang 050022 Hebei, China. [7]State Key Laboratory for Crop Genetics and Germplasm Enhancement and Utilization, CICMCP, Nanjing Agricultural University, Nanjing 210095 Jiangsu, China. [8]Hainan Yazhou Bay Seed Laboratory, Sanya, Hainan, China. [9]Centre of Excellence for Plant and Microbial Science (CEPAMS), JIC-CAS, Beijing, China. ✉e-mail: hqling@genetics.ac.cn; ljm@sjziam.ac.cn; jxiao@genetics.ac.cn

transporter)[4,5], and refining root architecture[6,7]. Nevertheless, further research is necessary to comprehensively understand the balance and coupling between root architecture response and nitrogen transporter expression, particularly when operating under low nitrogen conditions.

Kenong 9204 (KN9204) and Jing 411 (J411) are cultivars that exhibit diverse agricultural traits such as NUE, root architecture, and productivity under low nitrogen conditions[2]. By utilizing these superior materials, we successfully identified several NUE-related quantitative trait loci (QTLs), encompassing factors such as root length, root tips, and grain protein content, derived from recombinant isogenic lines (RILs) generated through a cross between KN9204 and J411 in previous studies[8–10]. We recently completed the reference genome sequence of KN9204 and identified 882 nitrogen metabolism genes (NMGs)[11]. Comparative transcriptome analysis between KN9204 and J411 revealed different responsive programs under low nitrogen constraint, especially in the yield-related spike tissue and in seeds during reproductive development[11]. The underground tissues (roots) are not as well investigated. Past research emphasizes the importance of the root architecture system (RSA) for NUE[12] and the significant differences of RSA that exist between KN9204 and J411[2,8]. However, the regulatory mechanism underlying the diverse transcriptional programs in the roots of KN9204 and J411 still remains to be elucidated.

Recent studies have emphasized the crucial role that epigenetic factors play in regulating both nutrition uptake and metabolism, as well as the synergistic plant response to nutrient availability[13]. For example, the *SET DOMAIN GROUP 8* gene is involved in regulating nitrogen (N) assimilation and lateral root response, by controlling the levels of H3K36me3 in response to changing nitrogen levels in *Arabidopsis*[14]. Similarly, the *HISTONE DEACETYLASE 19* gene is responsible for controlling root cell elongation and modulating the expression of phosphorus (Pi)-homeostasis genes in *Arabidopsis* under phosphate starvation conditions[15]. In rice, the polycomb repressive complex 2 (PRC2), which is recruited by NITROGEN-MEDIATED TILLER GROWTH RESPONSE 5, regulates tillering by depositing H3K27me3 on branching-inhibitor genes[16]. Additionally, the H3K27me3 level at the AtNRT2.1 (nitrate transporter) locus influences root nitrate uptake, a process that is mediated by HIGH NITROGEN INSENSITIVE 9 and PRC2[17]. Despite these advances in understanding the role of epigenetics in nutrient uptake and metabolism of plants like Arabidopsis and rice, the co-regulation of these processes and their response to nutrient availability in wheat remains largely unexplored.

In this work, we used the CUT&Tag[18,19] to profile epigenomic modifications in two wheat cultivars, KN9204 and J411, for various histone modifications and histone variant in three different tissues under varying nitrogen conditions. Our epigenomic maps offer a comprehensive understanding of the dynamic chromatin landscapes of these two wheat cultivars in response to low nitrogen condition. The disparity in chromatin modification levels between KN9204 and J411 cultivars influences the bias in nitrogen metabolism and has the potential to regulate gene expression in a cultivar-specific manner. Moreover, manipulating H3K27ac and H3K27me3 profiles through chemical inhibition or genetic mutation can significantly influence the LN adaptation strategy of KN9204 and J411.

## Results

### Profiling the tissue-specific chromatin landscape under different nitrogen conditions

To understand the epigenetic regulation of the transcriptomic dynamics (Supplementary Fig. 1a), we did CUT&Tag of various histone modifications for the wheat cultivars KN9204 and J411 at normal nitrogen (NN) and LN conditions for roots (28 days), flag leaves (heading stage) and seeds (21 days after anthesis) (Fig. 1a), corresponding to transcriptomes generated previously[11]. The effectiveness of the CUT&Tag assay in wheat was convincingly demonstrated

through its ability to unveil a robust correlation (Cor = 0.91/0.85) with previously published ChIP-seq data for H3K27ac[20] and H3K27me3[21], while also displaying good reproducibility between two biological replicates for different histone modifications (Supplementary Fig. 1b, c). The majority of histone modification peaks were located in the distal region expected for H3K36me3 in wheat (Supplementary Fig. 1d), as described in our recent report[19]. H3K27ac and H3K4me3 were associated with highly expressed genes, while H3K27me3 is enriched in low/non-expressed genes (Supplementary Fig. 1e). H3K4me3 and H2A.Z did not show a preference for gene expression level (Supplementary Fig. 1e). More than half of the H3K9me3 peaks were located in transposable elements (TEs) regions (Supplementary Fig. 1f). Genes involved in biotic and abiotic stress responses had a higher H2A.Z level relative to that in developmental and hormone-related genes (Supplementary Fig. 1g).

The chromatin state was systematically defined using ChromHMM[22], which integrated combinatorial patterns of various histone marks (Fig. 1b). Five major categories were formed from the fifteen chromatin states (CS) identified, including Promoter (CS1-4), Transcription (CS5-8), Enhancer-like (CS9-11), Repressive (CS12–14), and No signal (CS15), each with different genome coverage, TE enrichment, and gene expression level (Fig. 1b). Both the Promoter and Enhancer-like states were associated with H3K27ac, H3K4me3, and H3K27me3, but located in the transcription start site (TSS) and intergenic region, respectively (Fig. 1b). Repressive states were mainly enriched with H3K27me3, covering approximately 10% of the genome, while the no signal state accounted for a major portion (~83%) of the genome. Therefore, a limited portion of the genome (~7%) is transcribed or involved in transcriptional regulation in the context of various histone modifications (Fig. 1b).

We wonder how chromatin state dynamics reflect the differences between wheat cultivars, tissue types, and nitrogen conditions. For example, genes related to nitrate uptake in the root (*TaNRT2.3A*), assimilation in the leaf (*TaNIA_6D*), and amino acid storage in the seed (*TaPROT2_4D*) showed varied CS under LN condition in KN9204 (Fig. 1c). Generally, Enhancer-like CS was the most variable chromatin region between the different cultivars, while Promoter CS was primarily influenced by nitrogen availability (Fig. 1d). Furthermore, we calculated variability scores for different histone marks and found that H3K27ac and H3K27me3 exhibited the highest variability scores among the different nitrogen conditions and between the wheat cultivars (Fig. 1e). Further examination of the distribution patterns (Supplementary Fig. 1c) allowed us to categorize these histone marks into distal and promoter peaks and evaluate their variability using Pearson correlation analysis (Fig.1f, Supplementary Fig. 1h). We observed that distal H3K27ac showed cultivar-specificity, while promoter H3K27ac showed greater tissue-specificity (Fig. 1f). For H3K27me3, both distal and promoter peaks exhibited cultivar-specificities (Supplementary Fig. 1h). Therefore, distinct genomic regions with variable chromatin states are shaped by differences in wheat cultivars, tissue types, and nitrogen conditions, particularly with regards to H3K27ac and H3K27me3 marks.

### Cultivar-bias expression of NMGs is mainly mediated by histone modification variations

NMGs are crucial for plants to uptake and utilize nitrogen, including genes that encode nitrate transporter (NPF, NRT2, NAR2, CLC, and SLAH), nitrate reductase (NIA, NIR), ammonium transporter, and amino acid transporter (APC) family members, as well as enzymes involved in ammonium assimilation (GS, GOGAT, GDH, ASN, AspAT) and transcription factors (TFs) related to N metabolism[23,24]. Previously, we have identified a total of 882 NMGs in wheat through sequence similarity comparisons to six other plant species including *Brachypodium distachyon*, barley, rice, sorghum, maize, and *Arabidopsis*[11].

We compared the NMGs between KN9204 and J411 for variations in DNA variation, H3K27ac modifications in promoter regions, and

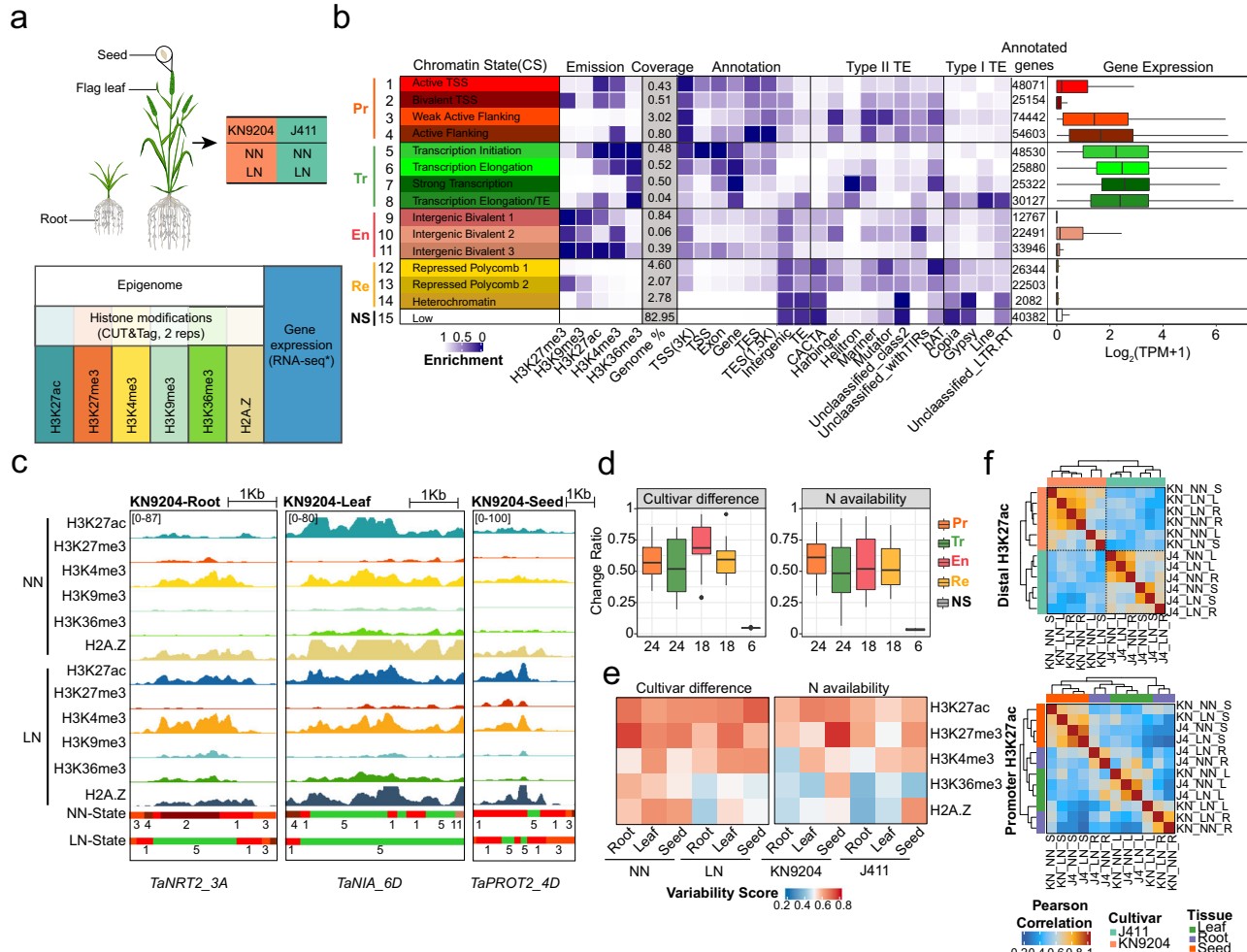

**Fig. 1 | Chromatin landscape dynamics in KN9204 and J411. a** Experimental design for generating the epigenomic datasets in KN9204 and J411 under different nitrogen conditions. RNA-seq raw data is from our previous publication[11]. **b** Chromatin states are depicted with different colors indicating distinct states: orange for promoter-related, green for transcription-related, pink for enhancer-like, yellow for repressed, and white for no signal. And genomic annotation enrichments were displayed, including emission, coverage, genome annotation, type II TE, and type I TE, alongside gene expression levels associated with each chromatin state. These states can be broadly categorized into five groups: Promoter (Pr), Transcription (Tr), Enhancer-like (En), Repressive (Re), and No Signal (Ns) states. Boxplots show the median, third, and first quartiles. The numbers indicate sample size used in the expression analysis. **c** Genome browser view of representative genes with dynamic chromatin states under different nitrogen conditions. The color code for the chromatin states are the same as in (**b**). NN, normal nitrogen; LN, low nitrogen. **d** The change ratio between different cultivars and nitrogen availabilities for the five categories of chromatin states. The numbers indicate sample size used in the analysis. Boxplots show the median, third and first quartiles. **e** Variability scores of different histone modifications among diverse tissues, wheat cultivars and nitrogen availabilities. **f** Cross-correlation heatmap of promoter and distal H3K27ac peaks. Abbreviations were as follow: KN KN9204; J4 J411, NN normal nitrogen, LN low nitrogen, S seed, R root, L flag leaf.

changes in transcriptional level. The results revealed that about 25% of NMGs displayed different levels of expression between KN9204 and J411 (FDR < 0.05, fold-change ≥ 1.5), while only around 5% of NMGs had DNA sequence variations in the regulatory regions (H3K27ac regions in promoter) (Fig. 2a). The variation in expression of NMGs between KN9204 and J411 existed for different gene families in various tissues (Fig. 2b, Supplementary Data 1). For instance, in roots, the expression levels of *NRT2* and *NIA* in J411 were both higher under the LN condition relative to KN9204, while the *NPF* family genes were activated in KN9204 (Supplementary Fig. 2a). Moreover, in flag leaves, *NIA* genes were up-regulated in KN9204, but not in J411. However, in seeds, the relative expression of genes that encode NPF and GS was much higher in KN9204 compared to J411 (Supplementary Fig. 2a).

The bias expression of NMGs in different cultivars is unlikely to be affected by variations in the DNA sequence within their regulatory regions (H3K27ac regions in promoter) as only 16 genes overlapped (Fig. 2c, top). Rather, the majority of NMGs that showed cultivar-biased expression were marked by differential peaks of H3K27ac (66.9%) and H3K27me3 (45.6%) (Fig. 2c, Supplementary Fig. 2b, Supplementary Data 2, Supplementary Data 3). Notably, we observed that members of the NRT2 and NIA families (comprising approximately 50% of genes marked by dynamic H3K27ac and H3K27me3 in the root) exhibited concurrent regulation by both H3K27ac and H3K27me3 (Supplementary Fig. 2c). In roots, the NRT2 family displayed similar changes in H3K27ac and H3K27me3 under LN/NN conditions in both cultivars but were more pronounced in J411 compared to KN9204 (Fig. 2d, top). For example, at the *TaNRT2_6A* (*TraesCS6A02G031000*) locus, H3K27ac gains and H3K27me3 losses in response to LN occurred in both cultivars but were more dramatic in J411 (Fig. 2e, top). This also aligns with the higher level of induced expression of *TaNRT2_6A* in J411 under LN conditions (Fig. 2e, top). *TaNPF2.3*, involved in nitrite transport from the root to the shoot[25], was activated in KN9204 with a decrease H3K27me3 under LN while it was accompanied by a significant decrease in H3K27ac in J411 (Fig. 2d, bottom). Consistently, *TaNPF2.3_7B* was increased in KN9204 but

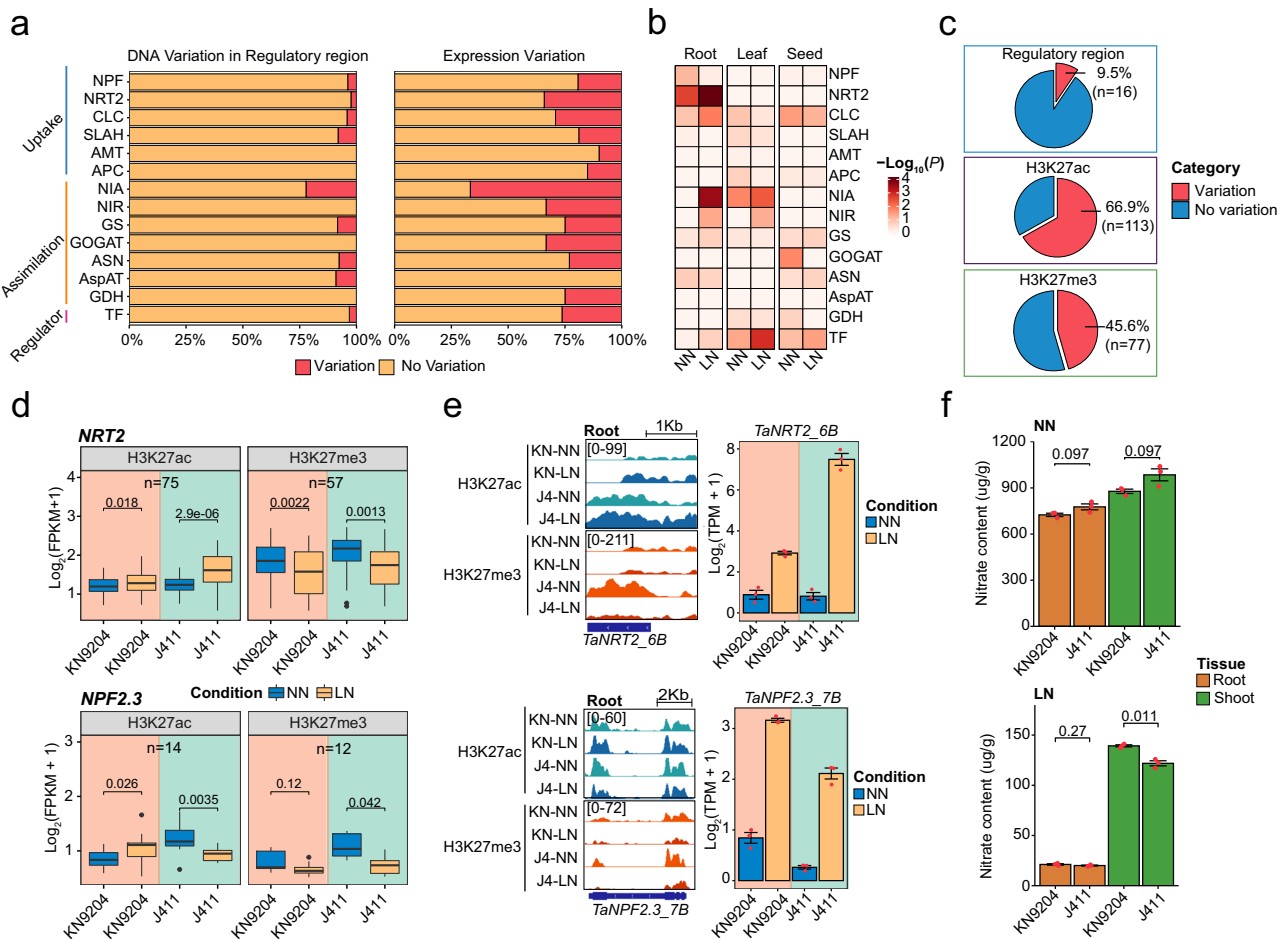

**Fig. 2 | Influence of epigenomic variations on the expression bias of NMGs.**
**a** Fraction of DNA variation in regulatory regions (promoter H3K27ac region) and expression variation of nitrogen metabolism genes (NMGs) between KN9204 and J411. Abbreviations: NPF NRT1/PTR FAMILY, NRT2 Nitrate transporter 2, CLC Chloride channel protein, SLAH Slow anion channel-associated homologs, AMT Ammonium transporter, APC The amino acid–polyamine–choline transporter superfamily, NIA Nitrate reductase, NIR Nitrite reductase, GS Glutamine synthetase, GOGAT Glutamate synthetase, ASN Asparagine synthetase, AspAT Aspartate aminotransferase, GDH Glutamate dehydrogenase, TF transcription factor.
**b** Enrichment of differentially-expressed NMGs between KN9204 and J411 based on their functional category in three tissues under different conditions (One-sided Fisher's exact test was used to calculate the *p* values for the overlaps). Abbreviations: same as (**a**). **c** Proportion of differential expressed NMGs with regulatory

region variation, H3K27ac variation, and H3K27me3 variation between KN9204 and J411. Red, variation; Blue, no variation. **d** Histone modification (H3K27ac and H3K27me3) levels of *NRT2* and *NPF2.3* genes in the roots of KN9204 and J411under two nitrogen availability levels (two-sided Wilcox test). Different shades indicate KN9204 and J411, respectively. The numbers indicate number of peaks used in the analysis. Box plots show the median, third, and first quartiles. **e** Genome browser view of histone modifications and bar graph view of transcriptional changes of *TaNRT2_6B* and *TaNPF2.3_7B* in roots for the two wheat cultivars and two nitrogen levels. Different shades indicate KN9204 and J411, respectively. Expression data shown as mean ± s.d. of three biological replicates. **f** Nitrate content of shoots and roots in KN9204 and J411 under two nitrogen conditions (two-sided Student's *t*-test). Data presented as mean ± s.d. of three biological replicates.

decreased in J411 under LN/NN conditions (Fig. 2e, bottom). Additionally, the nitrate contents in shoots relative to roots were higher in KN9204 compared to J411 under LN conditions rather than NN conditions (Fig. 2f). Similarly, cultivar-specific H3K27ac dynamic in response to LN was associated with expression change of ammonium assimilation enzymes coding genes *GS* and *GOGAT* in seeds (Supplementary Fig. 2d, e). Accordingly, KN9204 seeds have a higher protein content compared to J411 under LN conditions (Supplementary Fig. 2f). Taken together, the varied levels of H3K27ac and H3K27me3 were associated with the expression bias of NMGs, leading to different nitrogen metabolism processes in KN9204 and J411.

### Cultivar-specific H3K27ac influences NUE-related traits by transcriptional regulation
Given the importance of H3K27ac in regulating the chromatin state and expression patterns of NMGs across different tissues and cultivars at LN/NN, we extended to identify cultivar-specific H3K27ac regions

using K-means clustering (Fig. 3a). In general, the cultivar-specific H3K27ac peaks were primarily over-represented in distal regions while under-represented in promoter and genic regions when compared to all H3K27ac peaks (Fig. 3b). Moreover, cultivar-specific H3K27ac-marked promoters had close associations with cultivar-specific expressed genes in roots, leaves, and seeds (Supplementary Fig. 3a, b). For instance, H3K27ac-marked genes in KN9204 were associated with cell wall biogenesis and nutrient reservoir activity, such as *PROLINE TRANSPORTER 1* (*PROT1*), *LATERAL ORGAN BOUNDARIES-DOMAIN 16* (*LBD16*), XYLOGLUCAN ENDOTRANSGLUCOSYLASE/HYDROLASE 19 (*XTH19*), CELLULOSE-SYNTHASE-LIKE 5 (*CSLC5*) (Supplementary Fig. 3c), while H3K27ac specifically modulated genes involved in flavonoid biosynthesis in J411 (Supplementary Fig. 3c).

The cultivar-specific distal H3K27ac peaks were found to occupy the same genomic locations as H3K4me3 and H2A.Z (Supplementary Fig. 3d). Furthermore, these H3K27ac peaks exhibited a noteworthy enrichment of enhancer RNAs (eRNAs)[26] (Supplementary Fig. 3e),

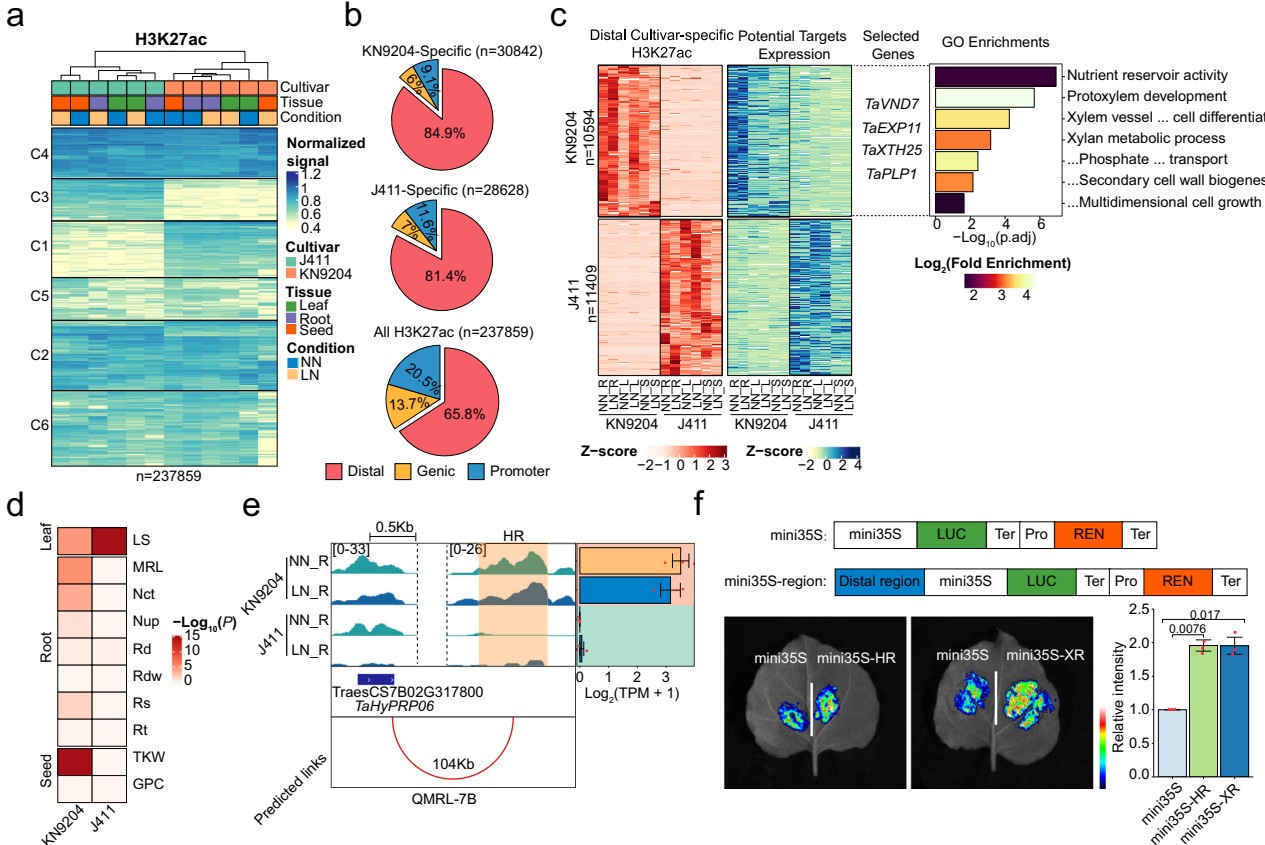

**Fig. 3 | Distal H3K27ac region divergence influences NUE related trait variation between KN9204 and J411. a** K-means clustering of variable H3K27ac peaks in three tissues of KN9204 and J411 under different nitrogen conditions. The numbers indicate number of peaks used in the analysis. **b** Distribution of cultivar-specific H3K27ac peaks and all H3K27ac peaks in the wheat genome categorized as distal, genic, or promoter regions, indicated by different colors. The numbers indicate number of peaks used in the analysis. **c** Assignment of cultivar-specific distal H3K27ac peaks to potential targets, with representative genes and GO enrichment displayed on the right (two-sided Fisher's exact test, BH for multiple comparisons). The numbers indicate number of distal peaks assigned to targets in the analysis. Abbreviations: NN_R NN_Root, NN_L NN_Leaf, NN_S NN_Seed, LN_R LN_Root, LN_L LN_Leaf, LN_S LN_Seed. **d** Enrichment analysis of cultivar-specific distal H3K27ac targets within QTLs between KN9204 and J411 (One-sided Fisher's exact test was

used to calculate the *p* values). Abbreviation: Rt root tip number, Rd root diameter, Nup Nitrogen uptake content, Nct Nitrogen concentration, MRL Maximum root length, TKW Thousand-kernel weight, GPC Grain protein content, Rs Root surface area, Rdw Root dry weight, Ls flag leaf size. **e** Representative tracks illustrating the regulation of *TaHyPRP06* by cultivar-specific distal H3K27ac peaks within *qMRL-7B*. The distal regulatory regions of *TaHyPRP06* was denoted as "HR". Abbreviations: NN_R NN_Root, LN_R LN_Root. Expression data shown as mean ± s.d. of three biological replicates. **f** Evaluation of the transcriptional regulatory role of cultivar-specific distal H3K27ac regions ("HR" and "XR") through luciferase reporter assay in the tracks shown in (**e**) and Supplementary Fig. 4e (two-sided Student's *t* test). Data shown as mean ± s.d. of three biological replicates. HR distal regulatory regions of *TaHyPRP06*; XR distal regulatory regions of *TaXTH25*.

indicating that these epigenetically modified hotspots may serve as 'enhancers' in regulating gene expression[27–29]. Remarkably, these cultivar-specific H3K27ac regions exhibit higher frequency of DNA variations than other distal H3K27ac region and notably higher than the promoter H3K27ac regions between KN9204 and J411, which also exhibit higher variation frequency in TF binding motifs (Supplementary Fig. 3f, g). As well, distal H3K27ac regions display higher frequency of DNA variation compared to promoter H3K27ac regions in a broader panel of cultivars[30,31] (Supplementary Fig. 3h). A correlation between the distal H3K27ac and gene expression dynamics allowed us to assign these regions to potential targets within a 500 kb distance as reported previously[32,33] (Supplementary Fig. 4a). A total of 58,493 unique pairs (22,003 distal H3K27ac regions and 6357 target genes) between the distal cultivar-specific H3K27ac peaks and genes were identified (Fig. 3c, Supplementary Data 4), and the accuracy of the pairs was supported by a higher chromatin interaction ratio from published Hi-C data[34] (Supplementary Fig. 4b). GO annotation of KN9204-specific distal H3K27ac-regulated genes showed enrichment in genes for cell wall synthesis, protoxylem development, and nutrient reservoir activity, which is similar to the KN9204-specific promoter H3K27ac-

marked genes. Among them, several genes are known to be involved in nutrient transport to the shoot, such as *VASCULAR RELATED NAC-DOMAIN PROTEIN 7* (*TaVND7*)[35], *XYLOGLUCAN ENDO-TRANSGLUCOSYLASE/HYDROLASE 25* (*TaXTH25*)[36]. No GO category was enriched in targets of J411-specific distal H3K27ac (Fig. 3c).

Differences in transcript abundance offer valuable insights into deciphering the genetic basis of QTLs, alongside variations in gene coding regions[37,38]. In this pursuit, we explored the impact of cultivar-specific distal H3K27ac regions on DEGs residing within QTLs derived from the KN9204-J411 RIL population[8–10]. In comparison to DNA sequence variations within promoter regions, cultivar-specific distal H3K27ac regions exhibited a more robust correlation with DEGs located within QTL regions associated with traits such as nitrogen uptake (Nup), nitrogen concentration (Nct), maximum root length (MRL), and grain protein content (GPC) (Supplementary Fig. 4c). There was a significant enrichment of target genes governed by KN9204-specific H3K27ac peaks within QTLs predominantly linked to NUE related-traits, where the elite genetic determinants hailed from KN9204 (Fig. 3d). Conversely, genes influenced by J411-specific H3K27ac regions were enriched in QTLs associated with leaf size-

related traits (Fig. 3d). For instance, within the well-documented qMRL-7B locus[8], we identified a total of 69 DEGs, with a mere nine genes exhibiting DNA sequence variations within their promoter regions. In contrast, 34 genes were marked by cultivar-specific distal H3K27ac peaks (Supplementary Fig. 4d, Supplementary Data 5). We observed significantly elevated expression levels of *TraesCS7B02G317800* (*hybrid proline-rich proteins 06, TaHyPRP06_7B*) and *TraesCS7B02G326900* (*TaXTH25_7B*), known regulators of root development[39,40], in KN9204 compared to J411. This heightened expression was correlated with presence of KN9204-specific H3K27ac distal regulatory regions (Fig. 3e, Supplementary Fig. 4e). Furthermore, the functional potential of these cultivar-specific regulatory regions was established through a luciferase reporter assay[41] (Fig. 3f, Supplementary Fig. 4f). However, it is important to acknowledge the challenge of fully understanding the regulatory functions of these distal regions due to the limited availability of precise 3D genome data in wheat. Thus, both promoter and distal cultivar-specific H3K27ac

regions play a significant role in modulating the expression of genes underlying NUE-related traits.

## LN-induced H3K27ac dynamics affect divergent adaptive programs in KN9204 and J411

We further investigated the role of H3K27ac dynamics induced by LN in driving divergent adaptations between KN9204 and J411. H3K27ac shows varied dynamic patterns in different tissues under LN/NN conditions in KN9204 and J411 (Supplementary Fig. 5a). KN9204 showed subtle changes in H3K27ac levels in roots, but more pronounced changes in flag leaves and seeds. Whereas J411 exhibited a significant loss and relatively lower gains in root but minor changes in aboveground tissues (Supplementary Fig. 5a, b). We found that dynamic change of H3K27ac leads different responses to LN for KN9204 and J411 in root (Fig.4a). The loss of H3K27ac in proximal regions (promoter and genic regions) caused down-regulation of genes that function in auxin homeostasis, cytokinin metabolism, and hormone signaling in J411 under LN/NN conditions (Fig. 4a). The gain of H3K27ac

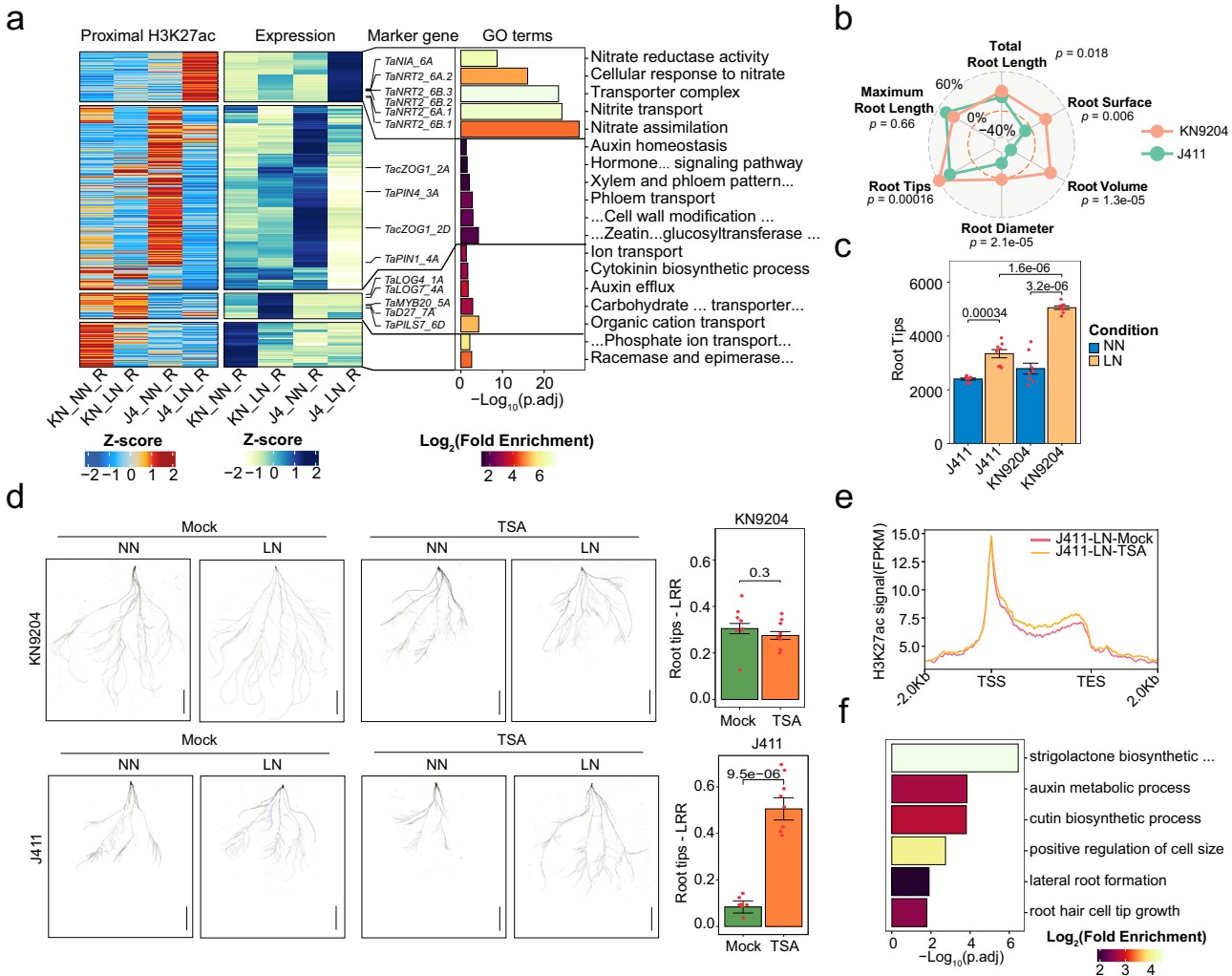

**Fig. 4 | Dynamics of H3K27ac revealing contrasting responses of KN9204 and J411 to low nitrogen conditions. a** Dynamic proximal H3K27ac peaks, corresponding gene expression changes, and GO (gene ontology) enrichments in the roots of KN9204 and J411. Enrichment significance assessed via two-sided Fisher's exact test. Abbreviations: KN_NN_R KN9204_NN_Root, KN_LN_R KN9204_LN _Root, J4_NN_R J411_NN_Root, J4_LN_R, J411_LN_Root. **b** The LN-response-ratio (LRR, details in Methods section) of root systems for KN9204 and J411(two-sided Student's *t* test). Data shown as the mean of eight biological replicates. **c** The number of root tips in KN9204 and J411 plants under different nitrogen conditions (two-sided

Student's *t* test). Data shown as mean ± s.d. of eight biological replicates. **d** Phenotypic characteristics of the root systems in KN9204 and J411 seedlings under control (mock) and TSA treatments (2 μM) in different nitrogen conditions. Scale bars = 5 cm. The right panel shows the LRR of the root tips, as defined in (**b**) (two-sided Student's *t*-test). Data shown as mean ± s.d. of eight biological replicates. **e** H3K27ac profiles of genes influenced by TSA treatment under LN condition in J411. **f** GO enrichment analysis of genes related to (**e**). Enrichment significance was assessed via two-sided Fisher's exact test.

is linked to the activation of genes responsible for nitrate uptake, transport and assimilation, including *TaNRT2* and *TaNIA* (Fig. 4a). In contrast, gain-of-H3K27ac in roots of KN9204 activated genes involved in cytokinin biosynthesis, auxin polarity transport, transporters for carbohydrate and organic cations, such as *DWARF27 (TaD27), PIN-LIKES 7 (TaPILS7)*, and *LONELY GUY 4 (TaLOG4)* (Fig. 4a), which function in root growth under LN as reported[42,43]. Loss of H3K27ac in KN9204 led to down-regulation of genes involved in phosphate ion transport. Consistently, the root system of J411 has a lower response to LN as compared to KN9204, with more root tips and larger root diameters under LN conditions (Fig. 4b, c). This eventually translated into an increase in root surface area and a larger total root volume in KN9204. Thus, the dynamic changes in H3K27ac that occur in response to LN preferably enhanced root growth in KN9204 while they strengthened the nitrogen uptake system in root of J411. In KN9204 flag leaves, increased H3K27ac were found in genes related to sucrose metabolism and xylem development, while J411 had more of such genes related to reactive oxygen species response (Supplementary Fig. 5c). In seeds, dynamic H3K27ac change have little influence on gene expression in both cultivars (Supplementary Fig. 5d).

Whether the root morphological change observed in KN9204 and J411 under LN/NN conditions is attributed to cultivar-specific H3K27ac alteration? To address this, we tested the effects of Trichostatin A (TSA), a non-specific chemical inhibitor of class I and II histone deacetylases[44], on H3K27ac pattern in wheat seedlings using a hydroponic culture system (details in "Methods" section). The effectiveness of TSA treatment was confirmed by western blotting (Supplementary Fig. 5e). Remarkably, TSA treatment was found to inhibit root growth under NN conditions. At the transcriptional level, stress response-related genes were up-regulated following TSA treatment (Supplementary Fig. 5f). Conversely, genes responsible for key enzymes involved in gibberellin biosynthesis, crucial for root development, were down-regulated, leading to root growth inhibition (Supplementary Fig. 5g). This phenomenon aligns with previous findings in *Populus trichocarpa*[45]. Phenotypically, root growth was induced by LN conditions in KN9204 in both the TSA-treated and untreated groups, evidenced by increased root tip numbers compared to NN conditions (Fig. 4d). In contrast, as expected, J411 did not display LN-induced root growth under mock conditions. However, when combined with TSA treatment, J411 exhibited a significant increase in the number of lateral root tips (Fig. 4d), rather than crown root tips (Supplementary Fig. 5h). Furthermore, our analysis identified specific genes in J411 where the reduction of H3K27ac under LN conditions was effectively counteracted by TSA treatment (Fig. 4e, Supplementary Fig. 5i). These genes are integral to the root development process (Fig. 4f) and were up-regulated under LN conditions (Supplementary Fig. 5j). This outcome underscores how TSA treatment restores higher expression levels of these genes, thereby contributing to enhanced root growth in J411 under LN conditions. In summary, TSA treatment was instrumental in mitigating the substantial LN-induced loss of H3K27ac in J411, ultimately restoring the root growth response of J411 to LN conditions.

## H3K27me3 shapes distinct root developmental programs in KN9204 and J411 under LN

In both KN9204 and J411, subtle changes in H3K27me3 occurred in flag leaves, while there were a large number of differential H3K27me3 regions in roots in J411 but in seeds in KN9204 (Supplementary Fig. 6a). K-means clustering further identified different categories of dynamic H3K27me3 regions in roots under LN conditions for KN9204 and J411 (Fig. 5a).

In root, significant changes occurred in J411 compared to KN9204 (Fig. 5a, Supplementary Fig. 6a). Additionally, genes marked with LN-induced gain-of-H3K27me3 (clusters C4, C5, and C7 in Fig. 5a) significantly overlapped with down-regulated genes in J411 (Fig. 5b, top). These genes were enriched in root growth-related processes, including

auxin biosynthesis, regulation of cell differentiation, and root morphogenesis (Fig. 5d, top). For example, the gene for a *Mob1-like* TF involved in root development[46] has increased H3K27me3 under LN conditions in J411 and the expression level is reduced, while no significant change in KN9204 (Fig. 5c, top). Conversely, the gain-of-H3K27me3 in KN9204 only overlapped with 24 genes that were down-regulated by LN (Supplementary Fig. 6b), implying that LN-induced gain-of-H3K27me3 in roots might mainly reduce the root growth process in J411 but not so much in KN9204. Loss-of-H3K27me3 in KN9204 and J411 produced significant overlaps with genes where expression increased in response to LN (Fig. 5b, middle and bottom). Genes involved in the response to nitrite, nitrite transport, and nitrate assimilation were enriched in J411, while genes involved in nitrate transport and SL and GA biosynthetic processes, as well as primary cell wall biogenesis-related genes, were enriched in KN9204 (Fig. 5d, middle and bottom). For example, expression of *TaNRT3.1_4A* and *ENT-KAURENOIC ACID HYDROXYLASE 2 (TaKAO2_4A)* was activated separately in J411 and KN9204 with decreased H3K27me3 (Fig. 5c, middle and bottom). Thus, The LN-induced H3K27me3 loss in roots tends to activate root growth in KN9204. Conversely, in J411, the loss of H3K27me3 activates nitrite uptake and metabolism. In the seeds, gain or loss of H3K27me3 in KN9204 did not lead too much change in overall gene expression (Supplementary Fig. 6c). The same is true for gain-of-H3K27me3 in J411 (Supplementary Fig. 6c). However, loss-of-H3K27me3 induced a significant amount of up-regulation of gene expression in J411, though no specific GO term was enriched (Supplementary Fig. 6c).

The deposition of H3K27me3 in plants depends on the recruitment of Polycomb repressive complex 2 (PRC2) by different DNA recognition factors[47,48]. To gain insights into the driving forces shaping the dynamic H3K27me3 landscape in two distinct plant varieties, KN9204 and J411, we undertook an in-depth motif scanning and enrichment analysis. This investigation unveiled intriguing disparities in the occurrence of specific motifs within KN9204 and J411-specific dynamic H3K27me3 regions, notably the "CGCCGCC" motif (accounting for 50%) and "GAGAGA" repeats (comprising 12.9%) (Fig. 5e). Remarkably, these motifs were also present in Polycomb response element (PRE) fragments in *Arabidopsis*[47,49]. Based on TF-DNA recognition in other species[50], ERF DOMAIN PROTEIN 9 (TaERF9_5B/ TaERF9_5D) and BASIC PENTACYSTEINE 1 (TaBPC1_4A) were selected for validation as potential recruiter of PRC2. Interestingly, *TaERF9* displayed differential expression patterns in response to LN conditions between KN9204 and J411. Specifically, it was induced to a higher degree in KN9204 under LN conditions (Supplementary Fig. 7a, b). On the other hand, *TaBPC1* exhibited higher expression level in J411 compared to KN9204 under LN conditions (Supplementary Fig. 7c). Additionally, we identified a single nucleotide polymorphism (SNP) within the exon of *TaERF9_5A*, resulting in an amino acid substitution (Lys-Glu) between KN9204 and J411 (Supplementary Fig. 7d). Using yeast two-hybrid (Y2H) and bimolecular fluorescence complementation (BiFC) assays, we verified that TaERF9_5B/TaERF9_5D (AP2/ERF family) could interact with TaEMF2 and TaSWN (subunits of PRC2) (Fig. 5f, g). Importantly, we found that TaBPC1_4A could interact with TaSWN (Figs. 5f, g, Supplementary Fig. 7e, f), aligning with previous research in *Arabidopsis*[51]. Thus, different TFs likely mediate LN-induced divergent H3K27me3 changes between KN9204 and J411, contributing to distinct root responses.

## Rewiring H3K27me3 modulates root development and nitrate uptake in response to LN

We investigated whether H3K27me3 plays a critical role in root development under LN conditions. There are nine genes encode H3K27me3 methyltransferases within three triads (Supplementary Fig. 8a), showing varied expressions in roots under different N conditions (Supplementary Fig. 8b). Considering the high expression of

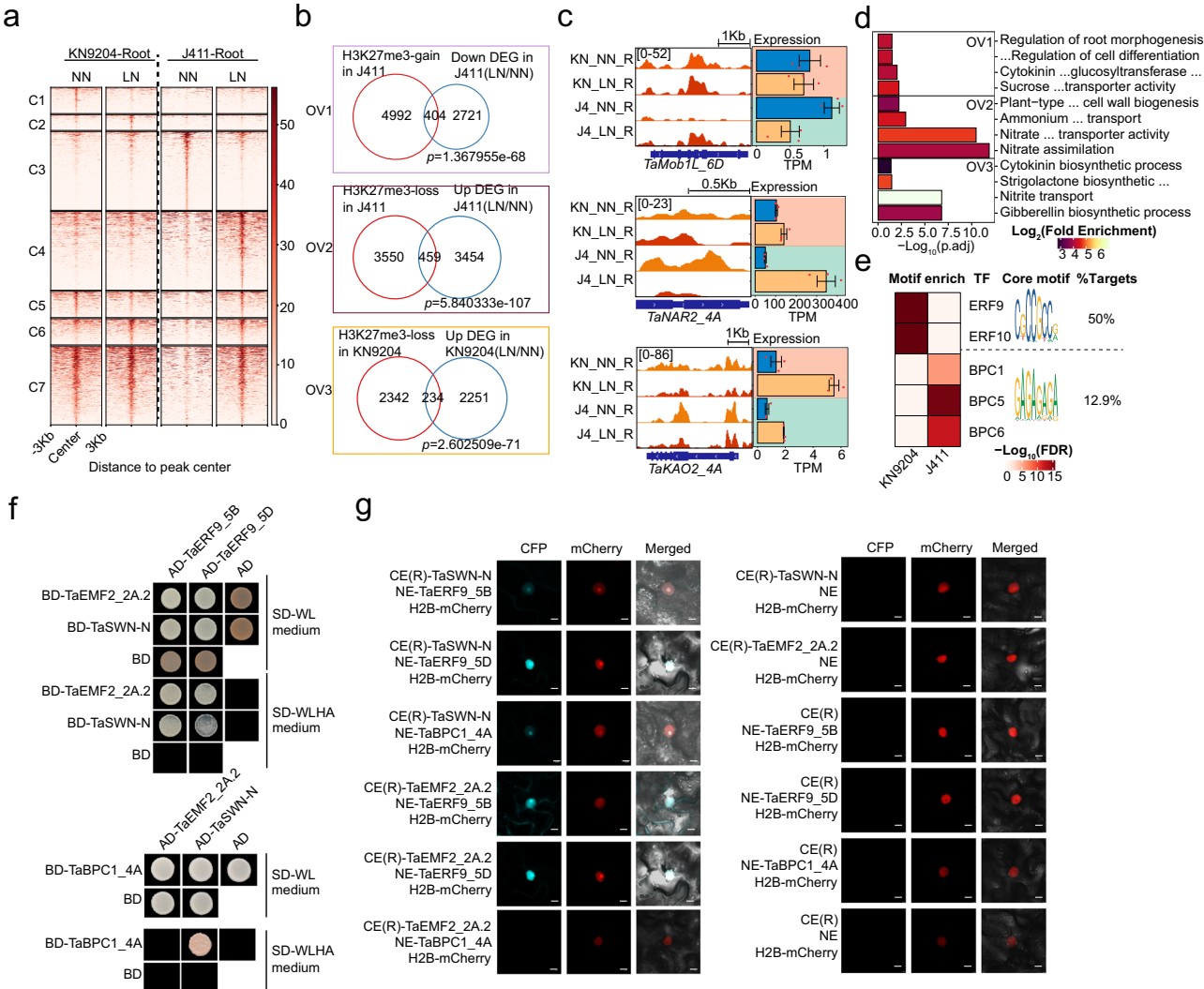

**Fig. 5 | Contrasting trends in H3K27me3 patterns shape differential LN adaptation in KN9204 and J411. a** Dynamic H3K27me3 peaks in the roots of KN9204 and J411 under LN and NN conditions. **b** Overlap between genes exhibiting dynamic changes H3K27me3 (H3K27me3-gain in J411, H3K27me3-loss in J411, and H3K27me3-loss in KN9204) and DEGs (down-regulated in J411, up-regulated in J411, and up-regulated in KN9204) in the roots. Significance was assessed using one-sided Fisher's exact test for the overlaps. OV1 reflect the transcriptional influence of LN-induced H3K27me3-gain in J411, OV2 reflects the transcriptional influence of LN-induced H3K27me3-loss in J411, OV3 reflect the transcriptional influence of LN-induced H3K27me3-loss in KN9204. **c** Representative tracks displaying H3K27me3 profiles and transcriptional alterations in *TaMob1L_6D*, *TaNRT3.1_4A*, and *TaKAO2_4A* for both cultivars under LN and NN conditions. Abbreviations: KN_NN_R KN9204_NN_Root, KN_LN_R KN9204_LN_Root, J4_NN_R J411_NN_Root, J4_LN_R

J411_LN_Root. **d** GO enrichment analysis of the overlapping genes mentioned in (**b**) (two-sided Fisher's exact test, BH for multiple comparisons), OV1-OV3 was same with (**b**). **e** Enrichment of sequence motifs for the up-regulated H3K27me3 peaks under LN conditions in KN9204 and J411. Significance assessed using one-sided Fisher's exact test for motif enrichment. **f** Yeast two-hybrid (Y2H) assays demonstrating the interaction between TaERF9_5B, TaERF9_5D, TaBPC1_4A and PRC2 components TaEMF2-2A.2, TaSWN-N (N-terminal of TaSWN). Transformed yeast cells were cultured on synthetic media lacking Leu and Trp (SD-WL) or Leu, Trp, His, and Ade (SD-WLHA). **g** Bimolecular fluorescence complementation (BiFC) analysis displaying the interaction between TaERF9_5B, TaERF9_5D, TaBPC1_4A and TaSWN-N, TaEMF2_2A.2. H2B-mCherry was used as control for transformation and localization of nuclei. Scale bars = 10 mm. CE(R), C terminal (right) of eYFP; NE, N terminal of eYFP. Experiment repeated two times with similar results.

*TaSWN* in roots and the limited genetic transformation efficiency of KN9204 or J411 (Supplementary Fig. 8b), we used the CRISPR-Cas9 system to create knock-out mutants of *SWINGER* (*TaSWN*, *TraesCS4A02G121300*, *TraesCS4B02G181400*, and *TraesCS4D02G184600*) in the amenable-to-transformation "Bobwhite" (BW) background (Supplementary Fig. 8c). Sequencing of transgenic wheat identified a *Taswn-cr* homozygous line with frameshift mutations in all three subgenome copies of *TaSWN* (Supplementary Fig. 8c). We then used CUT&Tag to compare the genome-wide H3K27me3 patterns in the *Taswn-cr* and BW. The peak number and length of H3K27me3 in *Taswn-cr* was significantly decreased compared to BW (Supplementary Fig. 8d). Similarly, the intensity of H3K27me3 on coding genes was decreased in *Taswn-cr* compared to BW (Supplementary Fig. 8e).

Therefore, TaSWN is indeed a H3K27me3 writer for some genomic areas in wheat.

Subsequently, we profiled H3K27me3 patterns in BW and *Taswn-cr* to see how TaSWN influences them under LN conditions. K-means clustering identified different categories of dynamic H3K27me3 regions in BW and *Taswn-cr* under either NN or LN conditions (Fig. 6a). Of note, the dynamic H3K27me3 peaks were predominantly located in distal regions, especially for clusters 1, 3, 6, and 7 (>80%) (Supplementary Fig. 8f). C1, C2, C6 and C8 clusters showed reduced H3K27me3 in *Taswn-cr* compared to BW under either LN or NN conditions (Fig. 6a), indicating TaSWN-dependent manner. We also found that about 18%-23% (*n* = 3057 and 19,683 respectively) of LN-induced H3K27me3 peaks in KN9204 or J411 were influenced by TaSWN

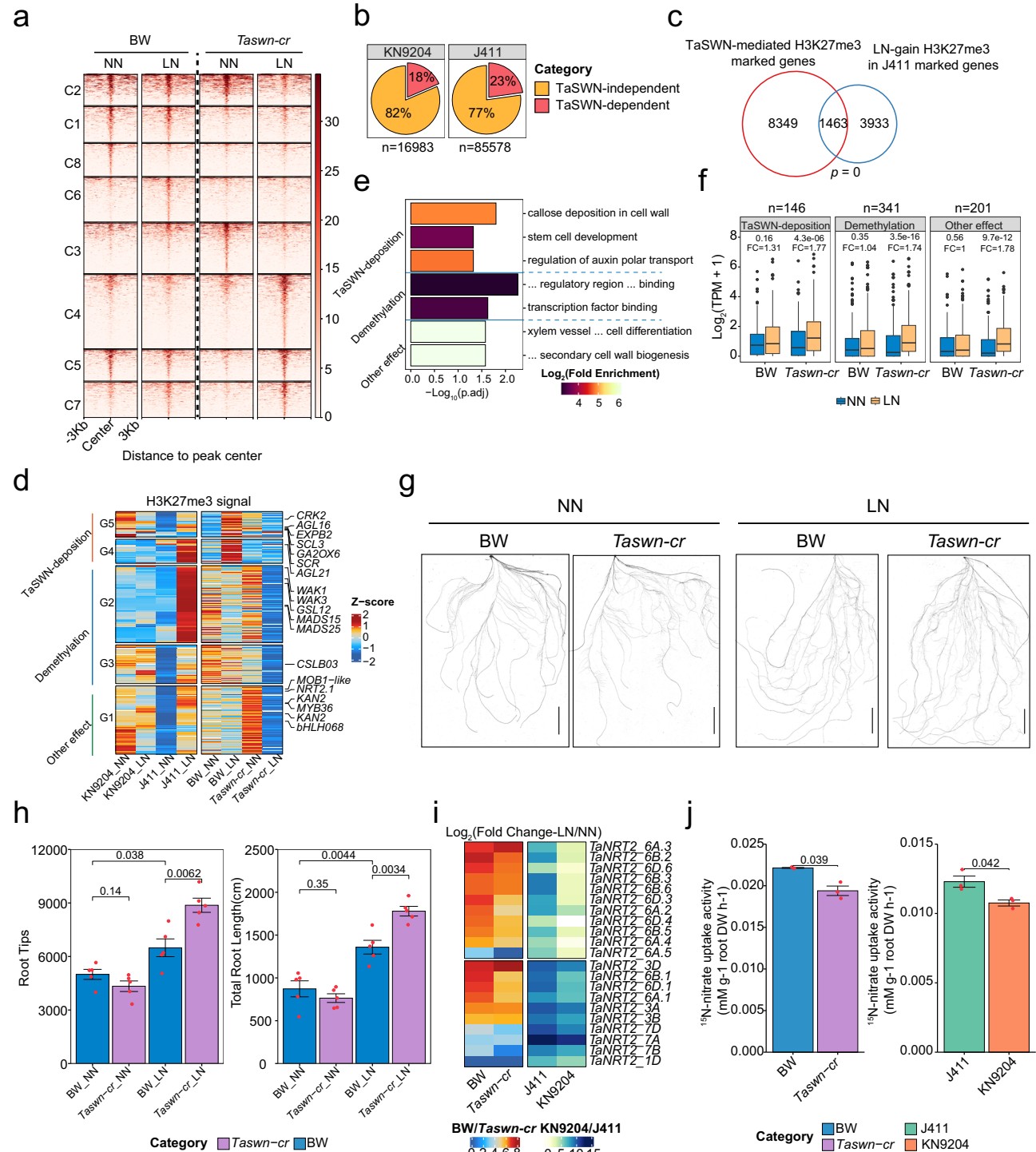

**Fig. 6 | Enhancing root adaptation of LN through H3K27me3 rewiring.**
**a** Dynamic H3K27me3 peaks in "BobWhite" (BW) and the *Taswn-cr* mutant plants under NN and LN conditions. **b** Proportion of TaSWN-dependent up-regulated H3K27me3 peaks induced by LN/NN conditions in KN9204 and J411. The numbers indicate total number of LN-induced H3K27me3 peaks. **c** Overlap between genes marked by TaSWN-dependent H3K27me3 and genes marked by LN-induced H3K27me3 in J411. Significance assessed using one-sided Fisher's exact test for overlaps. **d** Dynamic changes of H3K27me3 level influenced by TaSWN under different nitrogen conditions (NN and LN) in KN9204, J411, BW, and *Taswn-cr* plants. **e** GO enrichment of genes within different categories mentioned in (**d**) (two-sided Fisher's exact test, BH for multiple comparisons). **f** Expression changes and fold-

changes of genes within different categories mentioned in (**d**) (two-sided Wilcoxon test). FC: fold-change. Boxplots include a median with quartiles and outliers above the top whisker. **g** Scans of root systems of BW and *Taswn-cr* seedlings grown under different nitrogen conditions. Experiment repeated two times with similar results. Scale bars = 5 cm. **h** The number of root tips and total length of the root systems for BW and *Taswn-cr* seedlings grown under different nitrogen conditions (two-sided Student's *t*-test). Data are shown as mean ± s.d. of five biological replicates.
**i** Transcription fold-changes of *NRT2* genes under LN conditions in BW, *Taswn-cr*, J411, and KN9204 roots. **j** $^{15}$N-nitrate uptake activity of different seedlings (KN9204, J411, BW, and *Taswn-cr*) grown under LN conditions (two-sided Student's *t* test). Data are shown as mean ± s.d. of 3 biological replicates.

(Fig. 6b). J411 had more genes affected by SWN (*n* = 1463) compared to KN9204 (*n* = 155), suggesting H3K27me3 regulation is more significant in J411 (Fig. 6c, Supplementary Fig. 8g). Furthermore, we discovered specific gene sets with H3K27me3 changes driven by TaSWN in J411 under LN/NN conditions (Fig. 6d). For some genes (G4 and G5), TaSWN may directly added H3K27me3, while others (G2 and G3) saw LN-induced demethylation, especially in *Taswn-cr*, where the writer was lost. Genes in G1 had higher H3K27me3 in *Taswn-cr* under NN but dropped significantly under LN, with the cause unknown (Fig. 6d). These genes play roles in root development, affecting aspects like auxin transport, stem cell development, and transcription factors for root architecture formation (Fig. 6d, e). Furthermore, all subsets of genes exhibited elevated expression under LN/NN condition in *Taswn-cr* but no change in BW, aligning with the H3K27me3 reduction (Fig. 6f). Conversely, H3K27ac changes in *Taswn-cr* under LN/NN conditions mainly affected cytoskeleton processes and did not directly impact root development (Supplementary Fig. 8h, i). These findings suggest that TaSWN's influence on LN-induced H3K27me3 in J411 partly hinders root growth under LN conditions.

Therefore, we speculated that root growth in *Taswn-cr* plants would be more responsive to LN than BW. Indeed, we found roots were more developed in *Taswn-cr* compared to BW under LN, as determined by total root length and the number of root tips (Fig. 6g, h). In addition, the relative level of induction of *NRT2* expression in response to LN in *Taswn-cr* was lower than BW (Fig. 6i, left), which were similar to KN9204 cultivar with more developed roots (Fig. 6i, right). Consistently, the nitrate uptake rate was higher in BW compared to *Taswn-cr* as measured by $^{15}$N uptake assay under LN condition (Fig. 6j, left), which similar to the trend of J411 and KN9204 (Fig. 6j, right).

The comparison of root morphological changes, nitrate uptake rate and transcriptional profiles in response to LN between BW and *Taswn-cr* highlighted that H3K27me3 plays important role in balancing root growth and nitrogen metabolism under LN constraint. Rewiring H3K27me3 could influence wheat cultivars for the decision-making between significantly enhancing root growth or remarkably strengthening the nitrogen uptake system to adapt to low nitrogen environments.

## Discussion

As the urgency to reduce nitrogen fertilizer application in crop production, considerable effort has been directed towards dissecting the genetic basis of NUE regulation in crops[52–54]. However, epigenetic regulation, which functions in coordinating with TFs to manipulate gene expression, is not well studied in wheat. To fill this gap, we generated epigenomic datasets for three different tissues in two wheat cultivars that differ with respect to NUE related traits (KN9204 and J411) under different nitrogen conditions (Fig. 1). Our analysis revealed that the epigenome, which varies more than DNA sequence variation, plays an important role in mediating the cultivar-specific low nitrogen response.

Epigenetic modifications, especially H3K27me and H3K27ac, mediate transcriptional dynamics and contribute to different developmental programs or nitrogen metabolic processes between cultivars KN9204 and J411 (Figs. 1, 2). Bias-expressed NMGs are associated with altered epigenetic regulation patterns rather than DNA sequence variations between KN9204 and J411 (Fig. 2). Moreover, our investigation underscores that distal regions subject to epigenetic modifications distinctly reflect cultivar specificity (Fig. 1). Remarkably, these regions exhibit a higher frequency of DNA variations when compared to promoter regions, indicating their potential as a valuable resource for driving the formation of a wide array of traits (Supplementary Fig. 3).

In maize, many distal regulatory regions have been reported to regulate gene expression and are associated with agronomic trait variations[55,56]. Drawing parallels with findings in maize, genes

associated with cultivar-specific promoter H3K27ac or potentially regulated via the distal H3K27ac regions are enriched within QTL regions governing traits related to NUE, such as MRL, GPC, and Nup (Fig. 3). Those genes hold promise as candidates worth exploring for potentially mediate the genetic disparities between KN9204 and J411. Nonetheless, the biological significance of such epigenetic variation-mediated transcriptional regulation in shaping cultivar-specific agronomic traits necessitates validation through precision genome editing or even the use of epigenetic editing tools that targets specific distal regulatory regions or modification statuses. Looking ahead, profiling of the epigenome, particularly H3K27ac and H3K27me3, in a broader array of cultivars with well-defined NUE characteristics could pave the way for future Epi-GWAS analyses, ultimately unveiling the intricate regulatory mechanisms governing NUE.

To absorb adequate nitrogen in nitrogen-limiting conditions, cultivars with different agronomic features have various strategies, such as triggering root growth to have more root tips and increase the total root system volume in KN9204, or powering up the nitrate uptake machinery via up-regulation of NRT2 transporters in J411 (Fig. 7). Interestingly, the different adaptive strategies in roots between KN9204 and J411 are correlated with changes in the dynamic epigenome, especially for H3K27ac and H3K27me3 (Figs. 4, 5). Gain-of-H3K27ac and loss-of-H3K27me3 coordinately enhance the expression of root development-related genes in KN9204 under LN conditions, whereas loss-of-H3K27ac and gain-of-H3K27me3 reduce root development in J411 under LN, but rather activate nitrate uptake transporters via gain-of-H3K27ac and loss-of-H3K27me3 (Fig. 7). It is noteworthy that these two histone modifications, H3K27ac and H3K27me3, appear to target different sets of genes that influence root development (Supplementary Fig. 9). Several TFs show the potential to establish such epigenetic modification specificity via the recognition of certain cis-acting motifs and recruiting specific histone modification writers, such as ERFs and BPCs (Fig. 5). However, whether this precise modulation of epigenetic modifications, governing root development and nutrient absorption, represents a general mechanism for balancing plant growth and responding to environmental stimuli or adaptation across various cultivars remains a subject for broader investigation. Additionally, the intricacies of how these epigenetic modifications behave in a cultivar-specific manner remain largely unexplored. An intriguing discovery lies in the presence of potential histone writer or eraser-guiding TFs located within the QTL regions associated with traits like MRL and Nup (Supplementary Data 6). Further analysis promises to shed light on how LN triggers distinct response patterns in these TFs within KN9204 and J411.

In addition to the correlation between cultivar-specific epigenetic dynamics and varied strategies in response to LN, we have shown that manipulating the epigenetic features could affect the strategy selection for LN adaptation. Chemical inhibition (TSA treatment) or genetic manipulation (*Taswn-cr*) that changes the epigenome landscape (H3K27ac and H3K27me3) could lead to altered root system development and coordinated *NRT2* induction intensity under LN constraint (Figs. 4, 6). Similarly, several reports have shown that adjusting histone modification contributes to the response and/or tolerance to stress, such as HISTONE DEACETYLASE 6 (HDA6) regulated salt stress[57], and JUMONJI DOMAIN-CONTAINING PROTEIN 17 (JMJ17) regulated dehydration stress[58]. Interestingly, the enhancement of root growth is likely coupled with the attenuated induction of the expression of nitrate transporter coding genes under LN constraint by histone modification, in particular H3K27me3 (Fig. 7). Theoretically speaking, precise epigenomic modification alterations at specific regions/genes could help to decouple such linkage, which could generate wheat with developed root architecture system and higher induction of *NRT2* simultaneously. To achieve this, instead of the genetic manipulation of the "writer" or "eraser" to histone modification, fine-tuning of the driver (for example

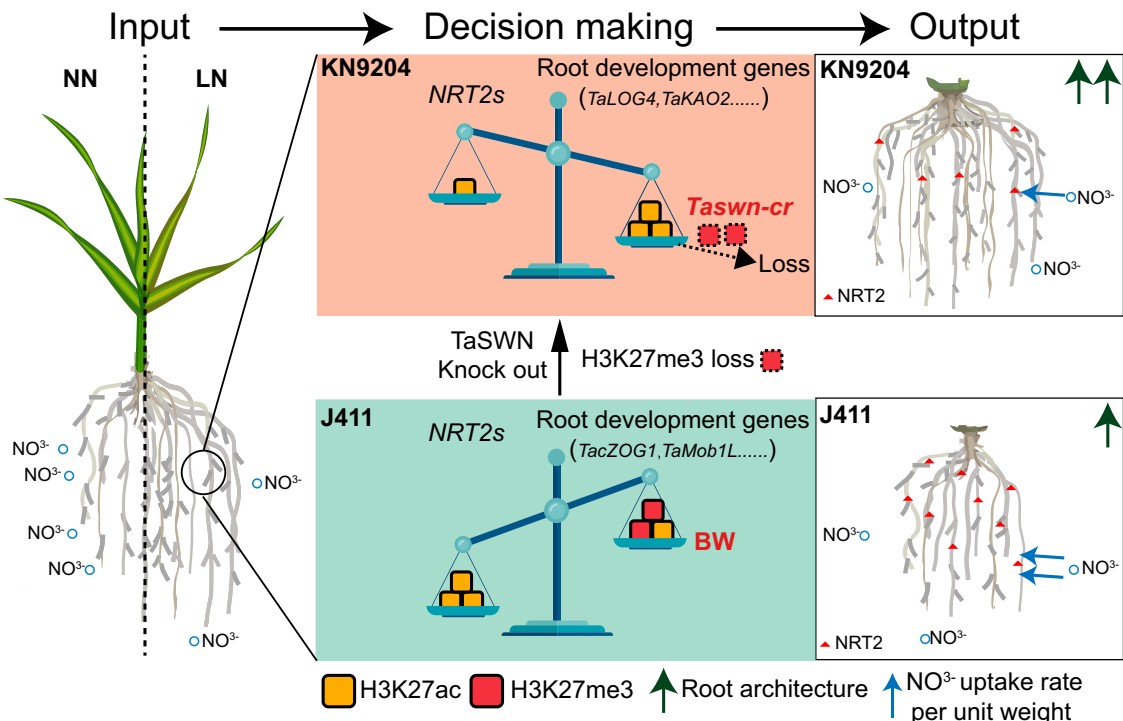

**Fig. 7 | A model of epigenetic regulation balancing diverse adaptation to LN in wheat.** In low-nitrogen condition (Input, left), different wheat cultivars select diverse adaptation strategies based on the equilibrium between root system development and nitrate uptake. Significant gain-of-H3K27ac enhance the expression of genes related to root development in KN9204. Conversely, a greater gain-of-H3K27me3 and a minor gain-of-H3K27ac reduce root development in J411 compared to KN9204, but with a simultaneous increase in nitrate uptake transporters

(*NRT2s*) via gain-of-H3K27ac. Upon the knock-out of *TaSWN* (*Taswn-cr*), root development is derepressed, accompanied by the loss of H3K27me3 (Decision making, middle). Phenotypically, KN9204 exhibits a well-developed root architecture (higher expression of root development genes) but a lower nitrate uptake rate per unit weight (lower expression of *NRT2s*), similar to *Taswn-cr*. In contrast, J411 exhibits a diverse selection regarding this balance, resembling BW (Output, right).

ERFs) may serve as a better way of coordinating nitrogen uptake and adaptive root growth to low nitrogen constraints in wheat cultivars.

## Methods

### Plant materials and culture conditions

The root of KN9204 and J411 was harvested 4 weeks after transplanting in the nutrient solution, which corresponding to 28-day in previous study[11], and immediately frozen in liquid nitrogen and stored at −80 °C. The nutrient solution for NN was as follows: 1 mM Ca(NO$_3$)$_2$, 0.2 mM KH$_2$PO$_4$, 0.5 mM MgSO$_4$, 1.5 mM KCl, 1.5 mM CaCl$_2$, 1 × 10$^{-3}$ mM H$_3$BO$_3$, 5 × 10$^{-5}$ mM (NH$_4$)$_6$Mo$_7$O$_{24}$, 5 × 10$^{-4}$ mM CuSO$_4$, 1 × 10$^{-3}$ mM ZnSO$_4$, 1 × 10$^{-3}$ mM MnSO$_4$, 0.1 mM Fe(III)−EDTA. For LN, 0.02 mM Ca(NO$_3$)$_2$, 2.48 mM CaCl$_2$ (to compensate for the Ca$^{2+}$ concentration in the nutrient solution), and other component was not changed. The flag leaf was harvested at heading stage[11] in the field (Shijiazhuang, China), and the seed was also harvested 21DAA[11] in the field (Shijiazhuang, China). In each NN plot, 300 kg/ha of diamine phosphate and 225 kg/ha of urea were applied before sowing, and 150 kg/ha of urea was applied at the elongation stage every year. In LN plots, no N fertilizer (N-deficient) was applied during the growing period.

### Generation of transgenic wheat plants

To obtain CRISPR transgenic wheat plants, the *pU6-gRNA* of *TaSWN* was annealed and inserted into *pJIT163-Ubi-Cas9* vector. All constructed vectors were transformed into callus to generate the transgenic plants.

To identify mutations in *TaSWN-4A*, *TaSWN-4B*, or *TaSWN-4D*, gene-specific primers were designed around the target site. Primers SWN-Check-F and SWN-Check-A-R were used to amplify *TaSWN-4A*, SWN-Check-F, and SWN-Check-B-R were used to amplify *TaSWN-4B*, and

SWN-Check-F and SWN-Check-D-R were used to amplify *TaSWN-4D*. PCR products were checked on agarose gels and genotyped by Sanger sequencing. Primer sequences were listed in Supplementary Data 7.

### RNA-seq and CUT&Tag experiment

Total RNA was extracted using HiPure Plant RNA Mini Kit according to the manufacturer's instructions (Magen, R4111-02), and libraries were sequenced using an Illumina Novaseq platform.

CUT&Tag experiment were done follow the previous described method[19]. The nuclei were extracted by chopping fresh samples soaked in the HBM buffer (25 mM Tris-HCl pH 7.6, 0.44 M sucrose, 10 mM MgCl$_2$, 0.1% Triton-X, 10 mM Beta-mercaptoethanol, 2 mM spermine, 1 mM PMSF, EDTA-free protease inhibitor cocktail). After overnight incubation with corresponding antibody in 4 °C, the nuclei were incubated in 50 μl wash buffer (20 mM HEPES pH 7.5; 150 mM NaCl; 0.5 mM Spermidine; 1× Protease inhibitor cocktail) with secondary antibody (1:100; Guinea Pig anti-Rabbit IgG antibody) at 4 °C for around 1–2 h and then washed twice with wash buffer. pA-Tn5 complex (pA-Tn5 1:100 dilution in CT-300 buffer:20 mM HEPES pH 7.5; 300 mM NaCl; 0.5 mM Spermidine; 1× Protease inhibitor cocktail) was incubated with nuclei in 4 °C for 2–3 h (Tn5, Vazyme, TD501-01). After washing twice with CT-300 buffer, the tagmentation of nuclei was done in 300 μl Tagmentation buffer (20 mM HEPES pH 7.5; 300 mM NaCl; 0.5 mM Spermidine; 1× Protease inhibitor cocktail; 10 mM MgCl2) in 37 °C for 1 h. 10 μl 0.5 M EDTA, 3 μl 10% SDS, and 2.5 μl 20 mg/ml Protease K were added to stop tagmentation reaction. The DNA was extracted with phenol:chloroform:isoamyl alcohol, precipitated with ethanol, and resuspended in ddH2O. The library was amplified 17 cycles by Q5 high-fidelity polymerase (NEB, M0491L). Antibodies (H3K27me3, CST, C36B11, 1/50 dilution; H3K4me3, Abcam, ab8582,

1/50 dilution; H3K27ac, Abcam, ab4729, 1/50 dilution; H3K36me3, Abcam, ab9050, 1/50 dilution; H3K9me3, Abcam, ab8898, 1/50 dilution; H2A.Z, Gift from Prof. Roger Deal lab, 1/50 dilution) used for histone modifications are the same as previous reported[19]. Libraries were purified with AMPure beads (Beckman, A63881) and sequenced using the Illumina Novaseq platform at Annoroad Gene Technology.

### $^{15}$N-nitrate uptake activity assay

$^{15}$N-nitrate uptake activity assay was performed as reported[59]. Wheat seedlings were grown in LN nutrient solution (0.1 mM $KNO_3$) for 28 days, respectively. After that, the seedlings were subjected to a pre-treatment of 3 h in LN nutrient solution (0.1 mM $KNO_3$). Subsequently, the seedings were transferred to LN (0.1 mM $KNO_3$ was replaced by 0.1 mM $^{15}$N-$KNO_3$) nutrient solution for $^{15}$N-labeling for a 3 h. Post $^{15}$N-labeling, the roots were washed using 0.1 mM $CaSO_4$ solution and deionized water. Finally, the shoots and roots of the seedlings were separately collected and dried at 70 °C until they reached a constant weight. Then, the samples were ground to fine powder and $^{15}$N-content was detected using an isotope ratio mass spectrometer (Isoprime 100).

### Nitrate ($NO_3^-$) content assay

Nitrate assay was performed as reported[60]. Standard curve was made based on different concentrations of $KNO_3$ solution (deionized water as a control). Samples were boiled at 100 °C for 20 min. After Centrifuge of the boiled different samples, salicylic acid-sulfuric acid was added to supernatant. After incubation for 20 min, 8% (w/v) NaOH solution was added. After cool down of the samples, measure the $OD_{410}$ value of each sample with the control for reference.

For samples collected of KN9204 and J411, grind each sample into powder in liquid nitrogen and conducted with same procedure as standard samples. Finally, calculate the nitrate concentration using the following equation: Y = CV/W.

*Y*: nitrate content (μg/g),
*C*: nitrate concentration calculated with $OD_{410}$ into a standard curve (μg/ml),
*V*: the total volume of extracted sample (ml),
*W*: weight of sample (g).

### Grain protein content assays

Grain protein content was measured by near-infrared reflectance spectroscopy (NIRS) with a Perten DA-7200 instrument (Perten Instruments, Huddinge, Sweden) and expressed on a 14% moisture basis. The measurements were calibrated using calibration samples according to the manufacturer's instructions.

### Luciferase reporter assay

The genomic sequence of distal regulatory region was amplified and fused in-frame with the *pMY155-mini35S* vector[41] to generate the reporter construct *cultivar-specific-regions-mini35Spro:LUC*. Then, *mini35Spro:LUC* (as control) and the reporter vector *cultivar-specific-regions -mini35Spro:LUC* were transformed into *A.tumefaciens* strain GV3101. The bacterial solution was injected to the back of the leaves of *Nicotiana benthamiana* (6–8 leaf stage) using a syringe with the needle removed. The *Nicotiana benthamiana* were cultivated for 2–3 days at a temperature of 22 °C and a light cycle of 16 h light/8 h dark. Firefly luciferase (LUC) and Renilla luciferase (REN) activities were measured using a dual luciferase assay reagent (Promega, VPE1910). And the relative intensity was calculated by ratio between relative ratio (LUC: REN) of cultivar-specific regions and empty vector. Primer sequences were listed in Supplementary Data 7.

### Root system scanning

The root of different samples was scanned by ScanMaker i8000 plus, after analyzed by WinRHIZO software, five root traits were quantified, including total root length (Rl), root surface area (Rs), root volume

(Rv), root diameter (Rd) and root tip number (Rt); and maximum root length is measured using a ruler.

LN response ratio (LRR) was calculated to reflect the root change under LN condition, which was ($R_{LN}$(root trait under LN condition) − $R_{NN}$(root trait under NN condition))/$R_{NN}$.

### TSA treatment

TSA (V900931-5MG) was dissolved in DMSO, then directly add into nutrient solution (NN/LN) to final concentration of 2 μM, with DMSO as mock. After treatment of 4 days, the root was harvested and immediately frozen in liquid nitrogen and stored at −80 °C.

### Protein interaction test (Y2H and BiFC)

For the Y2H between PRC2 and TaERF9_5B/ TaERF9_5D/TaBPC_4A, Full-length of *TaERF9_5B*, *TaERF9_5D* and *TaBPC1_4A* were amplified and fused with GAL4 AD in the *pDEST22* vector. Full length of *TaEMF2-2A.2* and N-terminal of *TaSWN* were amplified and fused with GAL4 BD in the *pDEST32* vector. Interactions in yeast were tested on the SD/-Trp/-Leu/-His/-Ade medium. And the interaction experiments between AtBPC1 and AtSWN/AtEMF2 in *Arabidopsis* was included as control. Primer sequences were listed in Supplementary Data 7.

For the BiFC analysis, the cDNA of *TaERF9_5B/ TaERF9_5D/ TaBPC_4A* and *TaEMF2-2A.2/TaSWN* was amplified and cloned into *pSCYCE* and *pSCYNE* vectors containing either C- or N-terminal portions of the enhanced cyan fluorescent protein. The resulting constructs were transformed into *A. tumefaciens* strain GV3101. Then, these strains were injected into tobacco leaves in different combinations with p19. The CFP fluorescence was observed with a confocal laser-scanning microscope (FluoView 1000, Olympus).

### Western blot assays

Total histone proteins were extracted by using EpiQuik Total Histone Extraction Kit (OP-0006-100). The total histone proteins were then used for western blot using the antibodies listed below. Anti-H3 immunoblot was used as a loading control. Antibodies: anti-H3 (ab1791, Abcam), anti-H3K27ac (ab4729, Abcam). Immunoblotting was done by using the enhanced chemiluminescence (ECL) system.

### RNA-seq data processing

Adapter sequence and low-quality reads of RNA-seq library were removed by fastp (0.20.1)[61], the cleaned reads were mapped to IWGSC Refseq v1.1 using hisat2 (2.1.0)[62], and gene expression was quantified by featureCount (2.0.1)[63]. Differentially expressed genes were evaluated using the DESeq2 package (1.34.0)[64] in R with an adjusted *p* value < 0.05 and log2 fold-change > 1. TPM (Transcripts Per Kilobase Million) values generated from the counts matrix were used to characterize gene expression and used for hierarchical clustering analysis.

For functional enrichment, GO annotation files were generated from IWGSC Annotation v1.1 and an R package clusterProfiler (4.2.2)[65] was used for enrichment analysis.

### CUT&Tag data processing

Adapter sequence and low-quality reads of CUT&Tag library were removed by fastp (0.20.1)[61], the cleaned reads were mapped to IWGSC Refseq v1.1 using bwa mem algorithm (0.7.17)[66], We further filter the reads mapped with "samtools view -bS -F 1,804 -f 2 -q 30" to filter the low-quality mapped reads. Then the high-quality mapped reads were reduplicated using Picard-2.20.5-0. The de-duplicated bam files from two biological replicates were merged by samtools (1.5)[67], and merged bam file was converted into bigwig files using bamCoverage provided by deeptools (3.3.0) with parameters "-bs 10 --effectiveGenomeSize 14,600,000,000 -- normalizeUsing RPKM --smoothLength 50". The bigwig files were visualized using deeptools (3.3.0)[68] and IGV (2.8.0.01)[69].

For peak calling, macs2 (2.1.4)[70] was used. For narrow peaks (H3K27ac, H3K4me3, and H2A.Z) and broad peaks (H3K27me3,

H3K36me3, and H3K9me3), parameters "-p 1e-3 --keep-dup all -g 14600000000" and "--keep-dup all -g 14600000000 --broad --broad-cutoff 0.05" were used. Peak was annotated to the wheat genome using the R package ChIPseeker (v1.30.3)[71], as peaks annotated to three categories: promoter (−3000bp of TSS), genic (TSS to TES) and distal (other). The MAnorm package[72] was used for the quantitative comparison of CUT&Tag signals between samples with the following criteria: |M value| > 1 and $P < 0.05$.

## Chromatin state analysis
For chromatin state analysis, chromHMM (1.21)[22] was used. "BinarizeBam" and "LearnModel" commands with default parameters were used for chromatin-state (CS) annotation. Multiple models were trained on these data, with CS numbers ranging from 2 to 20. The 15-state model was selected because it captured all the key information of CS. In previous studies, 15-state models were similarly trained on rice[73] and *Arabidopsis*[74] data.

For chromatin states dynamic change analysis, bins (CS called, 200 bp) were called dynamic if its state diverges between different samples. For variability score of histone modification, one minus jaccard index, which was calculated by bedtools (v2.29.2).

## Distal regulatory region-gene assignment
The distal regulatory regions annotation strategy was largely based on a previous study[33]. Genes within 0.5 M from a distal H3K27ac peak are considered candidate target genes. Then, we generated null model as correlations between randomly selected peaks and randomly selected genes on different chromosomes, and enabling us to compute mean and standard deviation of this null distribution. For each potential link, after calculate correlation between gene expression (TPM) and distal H3K27ac signal (FPKM) in samples, we also compute p-values for the test correlations based on null model, then significantly pairs were selected as regulatory region-gene pairs.

## Detection of transcription factor-binding motifs
To detect the recruiter of dynamic H3K27me3 changes, we downloaded the position weight matrices of plant motifs from the JASPAR database[50], the motifs were scanned by FIMO(4.11.2) within dynamic H3K27me3 regions. And enrichment test of motifs detected were done by fisher test in R.

## Reporting summary
Further information on research design is available in the Nature Portfolio Reporting Summary linked to this article.

## Data availability
The raw sequence data were deposited in the Genome Sequence Archive (https://bigd.big.ac.cn/gsa) under accession number CRA009936; the transcriptome used was under BioProject accession numbers CRA003969. Source data are provided with this paper.

## Code availability
Source code for analysis is available at https://github.com/ZhangHao-995/NUE-pipeline https://doi.org/10.5281/zenodo.10183398.

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

## Acknowledgements

We thank Dr. Yijing Zhang from Fudan University for providing the *pMY155-mini35S* construct. This research was supported by National Key Research and Development Program of China (2021YFF1000401, 2021YFD1201500), the National Natural Sciences Foundation of China (U22A6009), the Strategic Priority Research Program of the Chinese Academy of Sciences (XDA24010104), China Agriculture Research System of MOF and MARA (CARS-03), Hebei Natural Science Foundation (C2022503003) and China Agriculture Research System of MOF and MARA (CARS-03).

## Author contributions

J.X. designed and supervised the research, J.X. H.Z. J.-M.L. H.-Q.L. wrote the manuscript. H.Z. performed CUT&Tag, RNA-seq, nitrate assay, TSA treatment, yeast two-hybrid assay and root scanning experiments; H.Z. and L.Z. performed data analysis. X.-Y.Z. performed plasmid construction, part of yeast two-hybrid assay, BiFC experiment, and luciferase reporter assay. F.C. performed the determination of grain protein content. J.Z. performed Western blot; J.-C.C. performed plasmid construction for Taswn-cr. Z.-Y. J. and Y.-Y.L. performed $^{15}$N-nitrate uptake activity assay. C.-X.G. provided Taswn-cr transgenic wheat. H.Z. and J.X. prepared all the figures. Y.-P.L., Y.-X.N., L.W., W.-L.Z., X.-D.F., and Y.-P.T. polished the manuscript. All authors discussed the results and commented on the manuscript.

## Competing interests

The authors declare no competing interests.
