## [Peer Review File · Nature Communications]

REVIEWER COMMENTS

Reviewer #1 (Remarks to the Author):

This manuscript reports on dynamic epigenomic changes underlying distinct responses for adaptation to low nitrogen conditions between two wheat cultivars. The authors performed CUT&Tag to analyze six histone modifications/variants in root, leaf and seed. The authors found that the expression variations of nitrogen metabolism genes (NMGs) are more associated with epigenomic variations, especially H3K27ac and H3K27me3, rather than DNA variations. The authors revealed that dynamic changes in H3K27ac and H3K27me3 enhances root growth in KN9204, while strengthens the nitrogen uptake system in J411 in response to low nitrogen conditions. The manuscript provides insights on epigenetic regulation of cultivar-specific adaptation to limited nitrogen availability as well as epigenomic resources for the community. Although I found this study interesting, there are critical issues that need to be addressed. First of all, the authors need to show whether their CUT&Tag is well controlled by comparing with published CHIP-seq data. Although presented data clearly showed the enrichment of H3K27ac and H3K4me3 signals around TSS, the majority of peaks are located in distal intergenic regions. Since CUT&Tag is often highly biased to open chromatin regions, abundant co-localization of H3K27ac, H3K4me3 and H2A.z possibly reflect open chromatin regions rather than histone modification/variant distribution. Other comments are below.

1. The authors clearly showed dynamics of H3K27ac and H3K27me3. Since both marks occur at the same residue, they are mutually exclusive in the same cells. Both H3K27ac and H3K27me3 are associated with NMGs and root development related genes, there would be common target genes. How much are they overlapped? It would be helpful if both marks are presented side-by-side.

2. The authors identified predicted links between distal enhancer-like regions and gene expressions. The authors presented that one of such distal regions induced a reporter gene expression. As the authors stated “functional POTENTIAL of these cultivar-specific regulatory regions”, this does suggest the region potentially have regulatory function but it never guarantees its distal regulatory function. I would suggest to explicitly state the limitation of this analysis.

3. TSA treatment increased the number of root tips in J411 under LN conditions compared to mock treatment, suggesting that maintenance of H3K27ac levels suppressed the reduction in root tip number. However, which genes are responsible for this phenomenon? Also, why TSA treatment itself inhibits root growth? Additional transcriptome and epigenome analysis would clarify these questions and strengthen their conclusion. Did the authors count both crown root tip and lateral root tip? In addition, it would be better to state that TSA is a non-specific HDAC inhibitor.

4. The authors identified ERFs and BPCs as candidate TFs that recruit PRC2 complex to LN-induced H3K27me3 regions in KN9204 and J411, respectively. Unfortunately, the authors did not provide any mechanistic insight why distinct TFs are potentially responsible for PRC2 recruitment in each cultivar. At least, whether they are induced by LN in cultivar-specific manner should be examined. The authors showed physical interactions between these TFs and PRC2 complex factors EMF2 and SWN. However, BPCs physically interact with SWN but not with EMF2 in Arabidopsis. In this case, I would suggest to include the experiments using BPCs, SWN, EMFs of Arabidopsis as positive and negative controls.

5. The authors designated H3K27me3 gained in J411 under LN/NN conditions but lost in Taswn compared to BW as TaSWN-mediated LN-induced gain-of-H3K27me3. I understand clusters 1 and 4 in Fig6d fall into this category. However, although clusters 2,3,5 lost H3K27me3 in Taswn-cr-LN, they accumulated higher H3K27me3 in Taswn-cr-NN, indicating that H3K27me3 was demethylated by LN condition. In this situation, the authors cannot claim these regions are solely regulated by Taswn. Please discuss this. The number of root tips are associated with both H3K27ac and H3K27me3. Thus, H3K27ac analysis in Taswn mutants would be helpful.

Reviewer #2 (Remarks to the Author):

This is a substantial body of work describing involvement of epigenetics, specifically histone modifications, on wheat varietal-specific responses to N nutrition. It is based on a series of previously published studies of two key varieties differing substantially in yield (and therefore NUE directly as a metric) and in some aspects of low N responses (in root responses and N-metabolic gene expression). As a set of observations on an important aspect of gene regulation, namely the role of epigenetics, the manuscript has a lot of potential. However, whilst dealing with NUE as a central topic, no thought seems to have been given firstly to the underlying basis of the differences between the varieties, that is why they are differing in yield, and secondly consideration of whether the yield differences are the drivers for the NUE differences. Although there appear to be correlations between histone medication and specific gene expression relating to NUE, as published elsewhere (and cited here) for Arabidopsis, this is not the whole story and clearly there are more physiological and regulatory processes involved; some statements and some conclusions ought to be tempered appropriately.

The main issue is the density of presented data and the immensely complex figures with multiple panels and subpanels. Whilst such a thorough study is to be applauded there are some problems. The major issue is the lack of detail in the accompanying legends for almost all figures, which do not fully explain the figures and in a few cases are uninformative. For example, Fig 1c: the colours are not explained in 1b; reference in the discussion is made to Fig 1 for 'powering up the nitrate uptake machinery', which appears to be wrong. The lack of informative legends applies both to the main figures and the

supplementary figures. Overall, the figures need to be simplified with only the key finding included in the main paper.

Another issue is that quite wide-ranging conclusions are drawn from data on just two varieties. Is it correct to compare two such varieties (KN9204 and J411) which intrinsically have very different yield potentials? I have spent some time reviewing the previously published data from the authors on this material and I understand the investment by the authors in analysis of this material. Some efforts to seek the wider applicability of their hypotheses would be useful. However, on the contrary the GE work has used another variety (Bob White), when in fact for consistency, such an approach might have been appropriate using KN9204 and J411.

Reviewer #3 (Remarks to the Author):

Improving nitrogen use efficiency (NUE) is crucial for sustainable agriculture. KN9204 and J411 are different in nitrogen use efficiency (NUE), root architecture, and productivity under LN conditions. In this study, KN9204 and J411 were used to screen the histone modifications corresponding with the response to LN constraints. The authors established the relationship of epigenetic modification (H3K27ac and H3K27me3) with the LN adaptation strategy of different wheat cultivars.

MAIN POINTS:

1. It was concluded that cultivar-specific H3K27ac and H3K27me3 influences the bias expression of NMGs, which is unlikely affected by variation in the promoter sequence itself. Then, is the sequence different between KN9204 and J411 in distal regions? If it is different, could the bias expression be affected by sequence variation in distal regions?

Are distal regions or introns or UTRs included in “regulatory regions” mentioned in the paragraph 3 of RESULT Point 2?

“‘CGCCGCC’ motif (50%) and ‘GAGAGA’ repeat (12.9%) were enriched in KN9204- and J411-specific dynamic H3K27me3 regions, respectively.” It should be clarified whether the ‘CGCCGCC’ is unique motif for KN9204, and ‘GAGAGA’ is unique repeat for J411. The analysis of sequence variation in distal regions would help to search the sequence cause for the distal H3K27ac and NUE related-traits.

2. One QTL for a trait may contain dozens or even thousands of genes. Not all of these genes, however, control the trait. In general, a QTL contains only one or several genes that actually control the trait. For

example, qMRL-7B included 1,245 in the present study. It should be that one or several genes regulate the MRL. All of the 1,245 genes were analyzed in this study and the conclusion should be debatable.

3. “Furthermore, we confirmed the functional potential of these cultivar-specific regulatory regions through a luciferase reporter assay (Fig.3h, Fig. S3i).” The transcriptional regulation from epigenetic modification of the distal region should be validation.

4. “TaERF9_5B/TaERF9_5D and TaBPC_4A were selected for interaction tests with PRC2 components”. TaBPC_4A had been reported to interact with SWN and influence root development in Arabidopsis. Were TaERF9_5B/TaERF9_5D and TaBPC_4A found as the DEG or sequece-variant between KN9204 and J411?

4. “TaERF9_5B/TaERF9_5D and TaBPC_4A were selected for interaction tests with PRC2 components”. TaBPC_4A had been reported to interact with SWN and influence root development in Arabidopsis. Were TaERF9_5B/TaERF9_5D and TaBPC_4A found as the DEG or sequece-variant between KN9204 and J411?

5. “TaERF9_9B/TaERF9_5D (AP2/ERF family)”, “9B” should be “5B”

6. In “TraesCS7B02G317800 (TaHyPRP06_6B)”, “6B” should be “7B”.

REVIEWER COMMENTS

Reviewer #1 (Remarks to the Author):

This manuscript reports on dynamic epigenomic changes underlying distinct responses for adaptation to low nitrogen conditions between two wheat cultivars. The authors performed CUT&Tag to analyze six histone modifications/variants in root, leaf and seed. The authors found that the expression variations of nitrogen metabolism genes (NMGs) are more associated with epigenomic variations, especially H3K27ac and H3K27me3, rather than DNA variations. The authors revealed that dynamic changes in H3K27ac and H3K27me3 enhances root growth in KN9204, while strengthens the nitrogen uptake system in J411 in response to low nitrogen conditions. The manuscript provides insights on epigenetic regulation of cultivar-specific adaptation to limited nitrogen availability as well as epigenomic resources for the community.

Response: Thanks for the encouraging comments on our work.

Although I found this study interesting, there are critical issues that need to be addressed. First of all, the authors need to show whether their CUT&Tag is well controlled by comparing with published ChIP-seq data. Although presented data clearly showed the enrichment of H3K27ac and H3K4me3 signals around TSS, the majority of peaks are located in distal intergenic regions. Since CUT&Tag is often highly biased to open chromatin regions, abundant co-localization of H3K27ac, H3K4me3 and H2A.z possibly reflect open chromatin regions rather than histone modification/variant distribution.

Response: Thank you for the comments. We have added data analysis listed below:

1) There is limited published ChIP-seq data for root, flag leaf and seed tissues of wheat, as we sampled in this study. For comparison, we do seedling CUT&Tag for H3K27me3 and H3K27ac to compare with published ChIP-seq data of H3K27me3¹ and H3K27ac² from Chinese Spring seedling. We observed a strong correlation (H3K27me3: 0.85, H3K27ac: 0.91) (following figure a, Revised Fig. S1a) and similar peak distribution patterns (following figure b). Also the profiling of different histone modifications in the gene locus was also similar to results previously reported³ (following figure c). We also found good consistency in the specific sites (following figure d). This comparison supports that CUT&Tag works properly in our hands.

2) As reported previously³, ChIPseq of active histone mark H3K27ac, H3K27me3 also show major distribution pattern in distal intergenic regions. In large genome as wheat, there are large proportions of open regions in the distal intergenic regions, as evidences by us previously via ATAC-seq⁴. In comparison with ATAC-seq, different histone modifications showed varied enrichment in the open chromatin regions (following figure e). Specifically, H3K27ac exhibited high enrichment, whereas H3K9me3 showed lowest levels, which suggests that though there might be bias for CUT&Tag to enrich the open chromatin regions, it still show specific distributions for individual histone modification and variant, as ChIPseq.

3) H3K27ac, H3K4me3 and H2A.Z are associated with active transcribe regions (data from published DNase-seq experiments in wheat³), which is generally more open than control regions in the genome (following figure f). Following this logic, the co-localization region of all three histone features are likely to enrich potential enhancers. Indeed, we found a higher level of enhancer RNAs

(eRNAs), which was detected by GRO-seq⁵ and pNET-seq⁵, in these regions (KN9204-Specific or J411-Specific regions) compared to background in the genome (following figure g, Revised Fig. S3e).

In general, we believe the CUT&Tag profiling could re-capture ChIPseq pattern and indicate the specific distribution pattern of individual histone modifications and variant. As well, the co-localized regions for H3K27ac, H3K4me3 and H2A.Z are likely behaving like enhancer, with higher chromatin openness.

Comparison between ChIP-seq and CUT&Tag

a. Genome-wide Pearson correlation of H3K27ac and H3K27me3 (with 10-kb bin size) between published ChIP-seq^{1,2} and CUT&Tag. b. Peak distribution of H3K27ac and H3K27me3 from published ChIP-seq^{1,2} and CUT&Tag. c. Profiling of histone modifications from published ChIP-seq^{1,2} and CUT&Tag. d. Representative tracks showing good consistency between published ChIP-seq³ and CUT&Tag. e. The proportion peaks overlapped with open chromatin of different histone modifications. f. The enrichment of open chromatin³ in cultivar-specific and promoter/distal H3K27ac peaks, random regions in genome was selected as control. g. The mean expression level of eRNA⁵ in cultivar-specific and promoter/distal H3K27ac peaks, random

regions in genome was selected as control.

Other comments are below.

1. The authors clearly showed dynamics of H3K27ac and H3K27me3. Since both marks occur at the same residue, they are mutually exclusive in the same cells. Both H3K27ac and H3K27me3 are associated with NMGs and root development related genes, there would be common target genes. How much are they overlapped? It would be helpful if both marks are presented side-by-side.

Response: Thank you for your suggestion. We have conducted an overlap analysis between dynamic H3K27ac and H3K27me3 marks at NMGs and genes related to root development. You can find this analysis in the results (lines 168-171 in the revised manuscript) and discussion section (lines 439-441 in the revised manuscript)

1) It's noteworthy that regions marked by dynamic H3K27ac tended to have either unchanged or low levels of H3K27me3, and vice versa (Revised Fig. S9a). At the peak level, we observed a relatively low overlap, approximately 10%, between dynamic H3K27ac and H3K27me3 marks (Revised Fig. S9b).

2) Regarding NMGs, specifically genes from the NRT2 and NIA families, which constitute around 50% of the genes marked by dynamic H3K27ac and H3K27me3 in the root, we found that these genes were co-regulated by both H3K27ac and H3K27me3 (Revised Fig. S2c).

3) Root development-related genes exhibited diverse regulation patterns, including H3K27ac loss in J411, H3K27ac gain in KN9204, and H3K27me3 gain in J411. Our overlap analysis revealed that genes marked by H3K27ac gain in KN9204 and H3K27me3 gain in J411 had relatively low overlap (with no enriched functional terms) (Revised Fig. S9c). Additionally, genes that overlapped between H3K27ac loss and H3K27me3 gain in J411 (583 genes) were associated with processes such as xylem differentiation, sucrose transport, and response stimulus processes, rather than the core root development process (Revised Fig. S9d, e).

In summary, our analysis indicates that H3K27ac and H3K27me3 marks regulate distinct sets of genes, influencing root development in different ways.

2. The authors identified predicted links between distal enhancer-like regions and gene expressions. The authors presented that one of such distal regions induced a reporter gene expression. As the authors stated "functional POTENTIAL of these cultivar-specific regulatory regions", this does suggest the region potentially have regulatory function but it never guarantees its distal regulatory function. I would suggest to explicitly state the limitation of this analysis.

Response: Thank you for bringing up this important point. In response to your suggestion, we have explicitly addressed the limitations of our analysis in the revised manuscript. We state, " However, it is important to acknowledge the challenge of fully understanding the regulatory functions of these distal regions due to the limited availability of precise 3D genome data in wheat " (lines 242-244 in the revised manuscript).

3. TSA treatment increased the number of root tips in J411 under LN conditions compared to mock treatment, suggesting that maintenance of H3K27ac levels suppressed the reduction in root tip number. However, which genes are responsible for this phenomenon? Also, why TSA treatment itself inhibits root growth? Additional transcriptome and epigenome analysis would clarify these

questions and strengthen their conclusion. Did the authors count both crown root tip and lateral root tip? In addition, it would be better to state that TSA is a non-specific HDAC inhibitor.

Response: Thank you for your insightful comment. In response to your queries, we have taken additional steps to strengthen our analysis, with RNAseq and CUT&Tag for samples with TSA treatment, which are now incorporated into the revised manuscript (lines 277-282, 285-290 in the revised manuscript).

1) **Identification of DEGs influenced by TSA:** To elucidate the genes responsible for the observed increase in root tips with TSA treatment under low nitrogen (LN) conditions, we conducted further analysis. Specifically, we identified differentially expressed genes (DEGs) influenced solely by TSA treatment (under normal nitrogen conditions). Our findings indicate that the up-regulated DEGs are associated with stress response pathways (Revised Fig. S5f). Additionally, genes involved in gibberellic acid (GA) biosynthesis, a pathway crucial for root development, were downregulated by both TSA treatment and LN stress in both KN9204 and J411 (Revised Fig. S5g). These results align with previous findings in *Populus trichocarpa*⁶.

2) **Counting of crown root numbers:** To provide clarity, we have also counted the number of crown roots. Our analysis reveals that the number of crown roots was reduced under both Mock and TSA treatment conditions in response to LN stress (Revised Fig. S5h)

3) **Addition of RNA-seq and CUT&Tag data:** By integrating RNA-seq and CUT&Tag data, we identified genes in which the reduced level of H3K27ac in the LN condition was rescued by TSA treatment in J411 (Revised Fig. 4e, Fig. S5i). Notably, these genes are involved in the root development process (Revised Fig. 4f) and were upregulated under LN conditions (Revised Fig. S5j). This finding suggests that TSA treatment maintains higher expression levels of these genes, contributing to improved root growth in the LN condition in J411, consistent with the phenotype shown in Fig. 4d.

4) **TSA specificity:** We appreciate your point, and we have added a statement clarifying that TSA is a non-specific histone deacetylase (HDAC) inhibitor for better contextualization.

In summary, these additional analyses and explanations have enhanced our understanding of the effects of TSA treatment, gene expression, and root growth, as well as clarified the nature of TSA as a non-specific HDAC inhibitor.

4. The authors identified ERFs and BPCs as candidate TFs that recruit PRC2 complex to LN-induced H3K27me3 regions in KN9204 and J411, respectively. Unfortunately, the authors did not provide any mechanistic insight why distinct TFs are potentially responsible for PRC2 recruitment in each cultivar. At least, whether they are induced by LN in cultivar-specific manner should be examined. The authors showed physical interactions between these TFs and PRC2 complex factors EMF2 and SWN. However, BPCs physically interact with SWN but not with EMF2 in Arabidopsis. In this case, I would suggest to include the experiments using BPCs, SWN, EMFs of Arabidopsis as positive and negative controls.

Response: We appreciate your constructive suggestion, and we have taken steps to provide more insights into the mechanistic differences in TFs' roles in PRC2 recruitment and to include appropriate controls for our experiments. These updates are now incorporated into the revised manuscript (lines 333-336, 339-342 in the revised manuscript).

1) **Expression analysis of ERF9 and BPC1:** To address the differences in TF induction by LN in each cultivar, we conducted expression analysis. We observed that ERF9 exhibited distinct expression patterns in response to LN stress in KN9204 and J411. Specifically, ERF9 was induced to a higher level in KN9204 under LN conditions (Revised Fig. S7a, S7b). On the other hand, BPC1 showed downregulated expression in both J411 and KN9204 in response to LN stress, with KN9204 showing a more significant downregulation. However, it's noteworthy that BPC1 was expressed at a higher level in J411 compared to KN9204 under LN conditions (Revised Fig. S7a, S7c).

2) **Validation of protein interactions:** We apologize for any confusion regarding the interaction between PRC2 and BPC1. To clarify this, we conducted new experiments and included controls from Arabidopsis to validate these interactions (Revised Fig. S7e, f). Our updated results clearly show that BPC1 interacts with SWN but not with EMF2, which aligns with the known interactions in Arabidopsis (Revised Fig. 5f, g). These controls from Arabidopsis provide a valuable reference point for understanding the interactions in our context.

5. The authors designated H3K27me3 gained in J411 under LN/NN conditions but lost in *Taswn* compared to BW as TaSWN-mediated LN-induced gain-of-H3K27me3. I understand clusters 1 and 4 in Fig6d fall into this category. However, although clusters 2,3,5 lost H3K27me3 in *Taswn-cr-LN*, they accumulated higher H3K27me3 in *Taswn-cr-NN*, indicating that H3K27me3 was demethylated by LN condition. In this situation, the authors cannot claim these regions are solely regulated by *Taswn*. Please discuss this. The number of root tips are associated with both H3K27ac and H3K27me3. Thus, H3K27ac analysis in *Taswn* mutants would be helpful.

Response: We appreciate your detailed comment and have taken steps to re-analyze and clarify the section related to H3K27me3 changes dependent on TaSWN. We have also added H3K27ac CUT&Tag for *Taswn-cr* mutant as suggested.

1) **H3K27me3 change patterns:** We re-identify TaSWN-dependent peaks, specifically C1, C2, C6, and C8, which exhibited decreased H3K27me3 in *Taswn-cr* compared to BW under either LN or NN conditions (Revised Fig. 6a). Furthermore, we discovered specific gene sets with H3K27me3 changes driven by TaSWN in J411 under LN/NN conditions. Specifically, H3K27me3 peaks in G4 and G5 may directly added by TaSWN, as their H3K27me3 was lost in *Taswn-cr* under LN condition (Revised Fig. 6d). The dynamic H3K27me3 pattern observed in G2 and G3, exhibit LN-induced demethylation, as the H3K27me3 was uninfluenced under NN condition but lost under LN condition in *Taswn-cr* (Revised Fig. 6d). This demethylation may be mediated by *JMJ13* or *JMJ32* (following figure). In contrast, the peaks in G1 with even with higher H3K27me3 levels under NN condition in *Taswn-cr*, may result from combined effects of other factors, such as other "writer" and "eraser" of H3K27me3 (Revised Fig. 6d). These genes including *SCR*, *AGL21*, and *CRK2*, These genes play roles in root development, affecting aspects like auxin transport, stem cell development, and transcription factors for root architecture formation (Revised Fig. 6e). Furthermore, the expression of these genes, known to influence root development, was upregulated in the *Taswn-cr* under LN conditions, consistent with the phenotype of the *Taswn-cr* mutant (Revised Fig. 6f). In light of these findings, we have revised the writing in this section to provide a clearer explanation of the direct and indirect effects of TaSWN on H3K27me3 changes and their impact on root development (lines 370-378 in the revised manuscript).

Expression pattern of demethylation enzymes of H3K27me3

2) **H3K27ac analysis in *Taswn-cr* mutant:** By detecting the H3K27ac levels in *Taswn-cr*, we found that the pattern of change in response to low nitrogen (LN) was altered compared to BW (Revised Fig. S8h). The increase in H3K27ac in *Taswn-cr* activated genes that function in cytoskeleton-related processes, rather than directly impacting root development (Revised Fig. S8i) (lines 378-380 in the revised manuscript).

Reviewer #2 (Remarks to the Author):

This is a substantial body of work describing involvement of epigenetics, specifically histone modifications, on wheat varietal-specific responses to N nutrition. It is based on a series of previously published studies of two key varieties differing substantially in yield (and therefore NUE directly as a metric) and in some aspects of low N responses (in root responses and N-metabolic gene expression). As a set of observations on an important aspect of gene regulation, namely the role of epigenetics, the manuscript has a lot of potential. However, whilst dealing with NUE as a central topic, no thought seems to have been given firstly to the underlying basis of the differences between the varieties, that is why they are differing in yield, and secondly consideration of whether the yield differences are the drivers for the NUE differences. Although there appear to be correlations between histone modification and specific gene expression relating to NUE, as published elsewhere (and cited here) for Arabidopsis, this is not the whole story and clearly there are more physiological and regulatory processes involved; some statements and some conclusions ought to be tempered appropriately.

Response: Thanks for the encouraging comments on our work. We have revised the conclusion in this manuscript.

1) We have changed the original statement “*both promoter and distal cultivar-specific H3K27ac regions play a significant role in transcriptional regulation and contain genetic variations that are responsible for NUE-related traits*” to “Thus, both promoter and distal cultivar-specific H3K27ac regions play a significant role in modulating the expression of genes underlying NUE-related traits (line 244-246 in the revised manuscript).

2) We have changed original statement in the discussion section “*Indeed, genes associated with cultivar-specific H3K27ac either in the promoter or in the distal region are enriched in QTL regions for mediating NUE-related agronomic traits, such as MRL, GPC, and Nup (Fig. 3), which could be good candidates for mediating the genetic difference between KN9204 and J411*” to “Drawing parallels with findings in maize, genes associated with cultivar-specific promoter H3K27ac or potentially regulated via the distal H3K27ac regions are enriched within QTL regions governing traits related to NUE, such as MRL, GPC, and Nup (Fig. 3). Those genes hold promise as candidates worth exploring for potentially mediate the genetic disparities between KN9204 and J411” (line 417-421 in the revised manuscript).

3) We also added the limitation of biological importance of cultivar-specific epigenetic variation in the discussion section “Nonetheless, the biological significance of such epigenetic variation-mediated transcriptional regulation in shaping cultivar-specific agronomic traits necessitates validation through precision genome editing or even the use of epigenetic editing tools that targets specific distal regulatory regions or modification statuses” (line 421-425 in the revised manuscript).

4) Considering only two cultivars were used in this study, we also added the limitation in discussion section “However, whether this precise modulation of epigenetic modifications, governing root development and nutrient absorption, represents a general mechanism for balancing plant growth and responding to environmental stimuli or adaptation across various cultivars remains a subject for broader investigation. Additionally, the intricacies of how these epigenetic modifications behave in a cultivar-specific manner remain largely unexplored” (line 443-448 in the revised manuscript).

5) We have change the subtitles in the discussion part (line 406 and 452 in the revised manuscript).

The main issue is the density of presented data and the immensely complex figures with multiple panels and subpanels. Whilst such a thorough study is to be applauded there are some problems. The major issue is the lack of detail in the accompanying legends for almost all figures, which do not fully explain the figures and in a few cases are uninformative. For example, Fig 1c: the colours are not explained in 1b; reference in the discussion is made to Fig 1 for ‘powering up the nitrate uptake machinery’, which appears to be wrong. The lack of informative legends applies both to the main figures and the supplementary figures. Overall, the figures need to be simplified with only the key finding included in the main paper.

Response: Thank you for your feedback, and we appreciate your constructive comments. We acknowledge the complexity of some of our figures and the lack of detail in the accompanying legends. We sincerely apologize for any confusion cause. In response to your feedback, we have taken the following steps:

1) Simplification of figures: We have revised our figures (e.g. Figs. 3,4,5, Figs. S3, S4) to simplify them, retaining only the key findings in the main paper. This should enhance the clarity and accessibility of the figures for readers. Details were in follow:

- a. We removed the panel about influence of proximal H3K27ac regions to expression into revised Fig.S3.
- b. We removed the diagram of cultivar H3K27ac regions influence on DEGs in qMRL_7B into revised Fig.S4.
- c. We removed the panel about global change of H3K27ac pattern into revised Fig.S5.
- d. We removed the panel about global change of H3K27me3 pattern into revised Fig.S6.
- e. We deleted the panel and corresponding text about diff-peaks of H3K27me3 in response to LN.

2) Detailed Legends: We have revised all figure legends to ensure they provide comprehensive explanations of the figures, including color codes and any references necessary for full comprehension.

Another issue is that quite wide-ranging conclusion are drawn from data on just two varieties. Is it correct to compare two such varieties (KN9204 and J411) which intrinsically have very different yield potentials? I have spent some time reviewing the previously published data from the authors on this material and I understand the investment by the authors in analysis of this material. Some efforts to seek the wider applicability of their hypotheses would be useful. However, on the contrary the GE work has used another variety (Bob White), when in fact for consistency, such an approach might have been appropriate using KN9204 and J411.

Response: We appreciate your constructive suggestion and would like to address the concerns regarding the use of only two wheat varieties and the choice of Bob White for gene expression work.

1) Variety selection and adjustment of conclusions: We fully acknowledge your valid concern regarding the potential limitations of drawing broad conclusions from data collected from just two wheat varieties. The primary objective of our study was to generate comprehensive epigenome datasets and pinpoint variety-specific epigenetic regions responsible for regulating

agronomic traits linked to nitrogen use efficiency (NUE). In this context, we utilized KN9204 and J411 as representative varieties due to their distinct NUE characteristics⁷⁻⁹. To address the need for a more comprehensive understanding of NUE variation, we are actively expanding our analysis to include a broader spectrum of wheat varieties with varying NUE traits. It is important to note that we have diligently revised our conclusions and conducted an in-depth discussion regarding these limitations in the manuscript (lines 425-428 in the revised manuscript). Furthermore, we are presently engaged in profiling the epigenomes of numerous wheat varieties drawn from a GWAS panel, although we are constrained by budget limitations to study a smaller population. This effort is aimed at fortifying the broader applicability of the hypotheses and insights gleaned from our current study.

2) Use of Bob White for transgenic wheat: We opted to use Bob White for the transgenic work for two key reasons:

a. Transformation Efficiency: The transformation efficiency of KN9204 and J411 was notably low, making it challenging to obtain positive genetic transformed plants for experimentation. Bob White, on the other hand, exhibited a higher transformation efficiency, facilitating the gene expression studies.

b. Validation of H3K27me3 Role: We also aimed to investigate the effectiveness of H3K27me3 in response to low nitrogen conditions across different wheat varieties. To assess this, we conducted tests using Bob White. The results successfully validated the general functional role of H3K27me3 in the response to low nitrogen conditions across various wheat varieties.

Reviewer #3 (Remarks to the Author):

Improving nitrogen use efficiency (NUE) is crucial for sustainable agriculture. KN9204 and J411 are difference in nitrogen use efficiency (NUE), root architecture, and productivity under LN conditions. In this study, KN9204 and J411 were used to screen the histone modifications corresponding with the response to LN constraints. The authors established the relationship of epigenetic modification (H3K27ac and H3K27me3) with the LN adaptation strategy of different wheat cultivars.

MAIN POINTS:

1. It was concluded that cultivar-specific H3K27ac and H3K27me3 influences the bias expression of NMGs, which is unlikely affected by variation in the promoter sequence itself. Then, is the sequence different between KN9204 and J411 in distal regions? If it is different, could the bias expression be affected by sequence variation in distal regions?

Are distal regions or introns or UTRs included in “regulatory regions” mentioned in the paragraph 3 of RESULT Point 2?

“‘CGCCGCC’ motif (50%) and ‘GAGAGA’ repeat (12.9%) were enriched in KN9204- and J411-specific dynamic H3K27me3 regions, respectively.” It should be clarified whether the ‘CGCCGCC’ is unique motif for KN9204, and ‘GAGAGA’ is unique repeat for J411. The analysis of sequence variation in distal regions would help to search the sequence cause for the distal H3K27ac and NUE related-traits.

Response: Thank you for your detailed comments and questions. We have conducted a thorough analysis to address these concerns and provide clarity on the points raised in the revision.

1) DNA sequence variations in distal regulatory regions: We have analyzed the DNA sequence variations in distal regulatory regions between KN9204 and J411. Our findings demonstrate that these cultivar-specific distal regulatory regions exhibit a relatively higher frequency of DNA variations compared to other distal regions marked by H3K27ac. Significantly, this variation frequency is notably higher than that observed in promoter H3K27ac regions (Revised Fig. S3f). Accordingly, the frequency of DNA variations caused TFs binding motifs alteration is higher in cultivar-specific H3K27ac region as compared to others (Revised Fig. S3g). This observation suggests a potential link between DNA variations within these cultivar-specific histone modification regions and the distinct distal H3K27ac patterns, which, in turn, could influence the expression of genes located within the QTL regions associated with NUE-related traits, aligning with the reviewer's earlier prediction. (line 205-209 in the revised manuscript)

Importantly, we extended our analysis to encompass a broader population^{10,11} of wheat varieties beyond KN9204 and J411 (Revised Fig. S3h). This broader examination unveiled that, at the population level, distal epigenetically modified regions indeed display a higher frequency of DNA variations compared to promoter H3K27ac regions. This insight suggests that, collectively, DNA variations within distal epigenetically modified regions may constitute a valuable resource for driving the formation of diverse traits in wheat.

Expanding on this concept, we posit that profiling the epigenome, particularly focusing on H3K27ac and H3K27me3, across a more extensive array of cultivars characterized by well-

defined NUE traits will substantially bolster the functional analysis of NUE variation through a comprehensive "Epi-GWAS" targeting distal regulatory regions. We have revised the text to clarify this point (line 425-428 in the revised manuscript).

2) **Definition of "Regulatory Regions" in the text:** We apologize for any confusion in our previous text. In the paragraph 3 of RESULT Point 2, "regulatory regions" specifically refers to H3K27ac regions in the promoter. Distal regions, introns, or untranslated regions (UTRs) were not included in this term. We have revised the text to clarify this point (line 157 in the revised manuscript).

3) **Specific motifs in dynamic H3K27me3 regions:** We acknowledge the need for clarification regarding the 'CGCCGCC' motif and the 'GAGAGA' repeat. These motifs are not unique to KN9204 or J411 but rather differ in occurrence between these varieties within the dynamic H3K27me3 regions. We have revised the text to provide this clarification (line 326-329 in the revised manuscript).

2. One QTL for a trait may contain dozens or even thousands of genes. Not all of these genes, however, control the trait. In general, a QTL contains only one or several genes that actually control the trait. For example, qMRL-7B included 1,245 in the present study. It should be that one or several genes regulate the MRL. All of the 1,245 genes were analyzed in this study and the conclusion should be debatable.

Response: Thank you for raising this point, and we appreciate your attention to detail. We acknowledge the need for clarification regarding the number of genes within a QTL and their potential role in controlling the trait.

1) **Role of genes within QTL regions:** It is correct that a single QTL for a trait may contain a large number of genes, but not all of these genes necessarily control the trait. In general, a QTL typically contains only one or a few genes that actively regulate the trait in question. We apologize for any confusion caused by the original statement in our manuscript.

In our investigation of the qMRL-7B region, we identified 77 genes with DNA sequence variations within their coding sequences, out of a total of 1,245 genes in this region. Among these genes, approximately 80% exhibited missense mutations between the KN9204 and J411 varieties (following figure a), making them candidates for potentially mediating the genetic effects of qMRL-7B.

Genes with DNA variation in qMRL-7B

a. The different mutation types of genes with sequence variation in qMRL-7B. b. The overlap between genes with sequence variation and DEGs in qMRL-7B.

In addition to DNA sequence variations within coding regions, it is important to note that changes in gene transcription represent intermediate steps in the chain of events leading from genetic perturbation to phenotypic variation¹². Previous research has demonstrated that differentially expressed genes (DEGs) located within QTL regions have the potential to influence trait differences, as observed in the case of nitrogen-deficiency tolerance in sorghum¹³ and rice¹⁴. Within the same qMRL-7B region, we discovered 69 differentially expressed genes (DEGs) that displayed a smaller overlap with genes exhibiting altered sequence (n=5) (following figure b). Thus, the conclusion should not be interpreted as implying that all of these genes directly regulate MRL. Instead, our analyses try to identify genes within QTL regions that exhibited either coding variation and/or differential expression between the KN9204 and J411 varieties.

2) Functional effect of cultivar-specific H3K27ac bias: We found that the proportion of DEGs influenced by cultivar-specific H3K27ac bias was higher than that influenced by DNA variation (Fig. Sxx). For example, we identified genes like TaHyPRP06_7B, which are regulated by distal H3K27ac peaks and could potentially be involved in controlling the MRL trait.

We have revised the relevant sections in the manuscript (lines 222-246 in the revised manuscript) to provide a clearer explanation of our strategy and findings.

3. “Furthermore, we confirmed the functional potential of these cultivar-specific regulatory regions through a luciferase reporter assay (Fig.3h, Fig. S3i).” The transcriptional regulation from epigenetic modification of the distal region should be validation.

Response: Thank you for your insightful comment, and we deeply appreciate your concerns regarding the validation of the functional potential associated with cultivar-specific epigenetic modifications in distal regulatory regions.

In an ideal scenario, we would indeed aim to showcase the impact of these epigenetic modifications on target gene expression by selectively erasing histone modifications within these distal regions. However, the current landscape presents a challenge due to the absence of suitable methods for constructing or erasing specific epigenetic modifications within plant systems¹⁵. While there have been notable successes in specifically adding or removing DNA methylation and H3K27me3 in *Arabidopsis*^{16,17}, achieving the same level of specificity for H3K27ac modification in plants remains elusive. This is primarily due to the non-specific enzymatic activity of histone acetyltransferases and histone deacetylases on histone tails^{18,19}. Furthermore, in the context of wheat research, there are additional challenges, including low transformation efficiency, limitations related to specific cultivars, and the substantial time investment, typically around six months, required to obtain a limited number of transgenic wheat plants²⁰. These practical hurdles make it particularly challenging to conduct such experiments in wheat.

An alternative approach could involve the use of a protoplast system, which allows for specific epigenetic modification through the CRISPR-dCas9 system²¹. However, it's essential to note that the generation of protoplasts can lead to alterations in the genome-wide chromatin modification status²². This poses difficulties in maintaining the endogenous histone modification pattern for specific regions, potentially reducing the significance of in vitro validation assays.

In light of these practical challenges, we have taken the initiative to provide an open and transparent acknowledgment of the current limitations of our analysis within the revised manuscript (lines 421-425 in the revised manuscript). We believe that this transparency regarding the experimental constraints is vital for a comprehensive understanding of the scope and implications of our research.

4. “TaERF9_5B/TaERF9_5D and TaBPC_4A were selected for interaction tests with PRC2 components”. TaBPC_4A had been reported to interact with SWN and influence root development in Arabidopsis. Were TaERF9_5B/TaERF9_5D and TaBPC_4A found as the DEG or sequece-variant between KN9204 and J411?

Response: Thank you for your question. We have examined the expression patterns of TaERF9_5B, TaERF9_5D, and TaBPC_4A in response to low nitrogen (LN) conditions in both KN9204 and J411.

TaERF9_5B and TaERF9_5D exhibit distinct expression patterns in response to LN conditions between KN9204 and J411. Specifically, ERF9 showed a higher level of induction in KN9204 under LN conditions compared to J411 (Revised Fig. S7a, S7b). In the case of TaBPC_4A, we observed differential expression in response to LN conditions as well. Both J411 and KN9204 exhibited downregulated expression under LN conditions, with KN9204 experiencing a more pronounced downregulation (Revised Fig. S7a). Notably, TaBPC_4A showed higher expression levels in J411 compared to KN9204 under LN conditions (Revised Fig. S7c) (line 333-336, 339-342 in revised manuscript).

These differential expression patterns of TaERF9_5B, TaERF9_5D, and TaBPC_4A in response to LN conditions suggest potential roles for these genes in the response to nitrogen availability, with variations between the two wheat varieties. These variations may contribute to the observed differences in nitrogen use efficiency.

5. “TaERF9_9B/TaERF9_5D (AP2/ERF family)”, “9B” should be “5B”

6. In “TraesCS7B02G317800 (TaHyPRP06_6B)”, “6B” should be “7B”.

Response: Thanks for the careful checking. We have corrected those typos.

References:

1. Ramírez-González, R. H. et al. The transcriptional landscape of polyploid wheat. *Science* 361, eaar6089 (2018).
2. Yuan, J. et al. Open chromatin interaction maps reveal functional regulatory elements and chromatin architecture variations during wheat evolution. *Genome Biology* 23 (2022). <https://doi.org/10.1186/s13059-022-02611-3>
3. Li, Z. et al. The bread wheat epigenomic map reveals distinct chromatin architectural and evolutionary features of functional genetic elements. *Genome Biology* 20, 139 (2019).
4. Zhao, L. et al. Dynamic chromatin regulatory programs during embryogenesis of hexaploid wheat. *Genome Biology* 24, 7 (2023). <https://doi.org/10.1186/s13059-022-02844-2>
5. Xie, Y. et al. Enhancer transcription detected in the nascent transcriptomic landscape of bread wheat. *Genome Biology* 23, 109 (2022).
6. Ma, X., Zhang, C., Zhang, B., Yang, C. & Li, S. Identification of genes regulated by histone acetylation during root development in *Populus trichocarpa*. *BMC Genomics* 17, 96 (2016).

<https://doi.org:10.1186/s12864-016-2407-x>

7. Fan, X. et al. Identification of QTL regions for seedling root traits and their effect on nitrogen use efficiency in wheat (*Triticum aestivum* L.). *Theoretical and Applied Genetics* 131, 2677-2698 (2018). <https://doi.org:10.1007/s00122-018-3183-6>
8. Zhao, C. et al. QTL for flag leaf size and their influence on yield-related traits in wheat. *Euphytica* 214, 209 (2018). <https://doi.org:10.1007/s10681-018-2288-y>
9. Cui, F. et al. QTL detection for wheat kernel size and quality and the responses of these traits to low nitrogen stress. *Theoretical and Applied Genetics* 129, 469-484 (2016). <https://doi.org:10.1007/s00122-015-2641-7>
10. Zhou, Y. et al. *Triticum* population sequencing provides insights into wheat adaptation. *Nature Genetics* 52, 1412-1422 (2020). <https://doi.org:10.1038/s41588-020-00722-w>
11. Hao, C. et al. Resequencing of 145 Landmark Cultivars Reveals Asymmetric Sub-genome Selection and Strong Founder Genotype Effects on Wheat Breeding in China. *Molecular Plant* 13, 1733-1751 (2020). <https://doi.org:10.1016/j.molp.2020.09.001>
12. Mackay, T. F. C., Stone, E. A. & Ayroles, J. F. The genetics of quantitative traits: challenges and prospects. *Nature Reviews Genetics* 10, 565-577 (2009). <https://doi.org:10.1038/nrg2612>
13. Li, Q. et al. Genome-wide association study and transcriptome analysis reveal new QTL and candidate genes for nitrogen-deficiency tolerance in rice. *The Crop Journal* 10, 942-951 (2022). <https://doi.org:10.1016/j.cj.2021.12.006>
14. Li, Q. et al. Genome-wide association study and transcriptome analysis reveal new QTL and candidate genes for nitrogen-deficiency tolerance in rice. *The Crop Journal* 10, 942-951 (2022). <https://doi.org:10.1016/j.cj.2021.12.006>
15. Gardiner, J., Ghoshal, B., Wang, M. & Jacobsen, S. E. CRISPR–Cas-mediated transcriptional control and epi-mutagenesis. *Plant Physiology* 188, 1811-1824 (2022). <https://doi.org:10.1093/plphys/kiac033>
16. Borg, M. et al. Targeted reprogramming of H3K27me3 resets epigenetic memory in plant paternal chromatin. *Nature Cell Biology* 22, 621-629 (2020). <https://doi.org:10.1038/s41556-020-0515-y>
17. Gallego-Bartolomé, J. et al. Targeted DNA demethylation of the *Arabidopsis* genome using the human TET1 catalytic domain. *Proceedings of the National Academy of Sciences* 115, E2125-E2134 (2018). <https://doi.org:10.1073/pnas.1716945115>
18. Wapenaar, H. & Dekker, F. J. Histone acetyltransferases: challenges in targeting bi-substrate enzymes. *Clinical Epigenetics* 8, 59 (2016). <https://doi.org:10.1186/s13148-016-0225-2>
19. Milazzo, G. et al. Histone Deacetylases (HDACs): Evolution, Specificity, Role in Transcriptional Complexes, and Pharmacological Actionability. *Genes* 11 (2020).
20. Wang, K., Liu, H., Du, L. & Ye, X. Generation of marker-free transgenic hexaploid wheat via an *Agrobacterium*-mediated co-transformation strategy in commercial Chinese wheat varieties. *Plant Biotechnology Journal* 15, 614-623 (2017). <https://doi.org:https://doi.org/10.1111/pbi.12660>
21. Gentzel, I. N. et al. A CRISPR/dCas9 toolkit for functional analysis of maize genes. *Plant Methods* 16, 133 (2020). <https://doi.org:10.1186/s13007-020-00675-5>
22. Xu, M., Du, Q., Tian, C., Wang, Y. & Jiao, Y. Stochastic gene expression drives mesophyll protoplast regeneration. *Science Advances* 7, eabg8466 <https://doi.org:10.1126/sciadv.abg8466>

REVIEWERS' COMMENTS

Reviewer #1 (Remarks to the Author):

The authors added new experiments/discussion and fully addressed my concerns.

Reviewer #2 (Remarks to the Author):

Thanks you for attending to my comments. The manuscript is much improved.

Reviewer #3 (Remarks to the Author):

The authors have carried out the reanalysis, and has made the reasonable revision in the revised manuscript. For the fifth question, the authors have given the reasonable reasons. I suggest that this manuscript should be acceptable.